



# Measurement of the turning behaviour of tethered membrane wings using automated flight manoeuvres

Christoph Elfert[1], Dietmar Göhlich[1], and Roland Schmehl[2]

[1]Methods for Product Development and Mechatronics, Technische Universität Berlin, 10623 Berlin, Germany
[2]Faculty of Aerospace Engineering, Delft University of Technology, 2629 HS Delft, Netherlands

**Correspondence:** Roland Schmehl (r.schmehl@tudelft.nl)

**Abstract.** Flexible membrane wings for kite sports, paragliding and airborne wind energy are highly manoeuvrable aerodynamic devices. The manoeuvrability can be quantified by the achievable turning rate of the wing and the dead time between the steering input and the actual flight-dynamic response. In this paper, we present an onboard sensor system for measuring the position and orientation of a tethered membrane wing and complement this with an attached low-cost multi-hole probe

for measuring the relative flow velocity vector at the wing. To ensure well-defined flow conditions and high quality of the measurement data, the wings selected for testing were towed by a vehicle with constant speed along a straight track during periods of low ambient wind speeds. A flight control algorithm was adapted from literature to execute automated, repeatable figure-of-eight flight manoeuvres and measure the steering gain and the dead time as functions of the steering input. The experimental study confirms the turning behaviour known from kite sports and airborne wind energy applications and provides

reproducible quantitative data to develop and validate simulation models for flexible, tethered membrane wings.

## 1  Introduction

Flexible membrane wings are commonly used for applications where mobility, low weight, low cost and fast deployment are essential. Prominent examples are kiteboarding, snowkiting, paragliding and airborne wind energy (AWE). The latter is an emerging renewable energy technology that is considered complementary to conventional wind energy because it can access

wind at higher altitudes (Bechtle et al., 2019; Kleidon, 2021) while using only a fraction of the material resources required for conventional wind turbines (IRENA, 2021). AWE systems based on flexible membrane wings are currently being developed by the companies Skysails, Kitenergy and Kitepower, all using pumping cycle operation to harvest wind energy (Nelson, 2019). The development of this specific wing type is still a largely iterative empirical process based on the experience of kite designers and subjective prototype tests.

Unlike a fixed wing, a flexible membrane wing does not have sufficient bending stiffness in span- and chord-wise directions and must be supported by a bridle line system. Instead of using individually actuated aerodynamic control surfaces, the entire wing functions as a morphing control surface, actuated by the tethers or the bridle line system. Asymmetric actuation is generally used for steering, while some designs also allow for symmetric actuation of the rear bridle lines for powering and depowering (Oehler and Schmehl, 2019). A notable exception is the ram air wings of Skysails, which, until recently, were





steered by roll control and not by wing morphing (Paulig et al., 2013). Because of the general C-shape of a bridled membrane wing, any changes to the geometry of the bridle line system also entail a deformation of the tensile membrane structure. In addition to this actuation-induced morphing, a membrane wing is also subject to aeroelastic deformations.

The spanwise twisting of leading-edge inflatable (LEI) kites during sharp turning manoeuvres is illustrated in Fig. 1. The

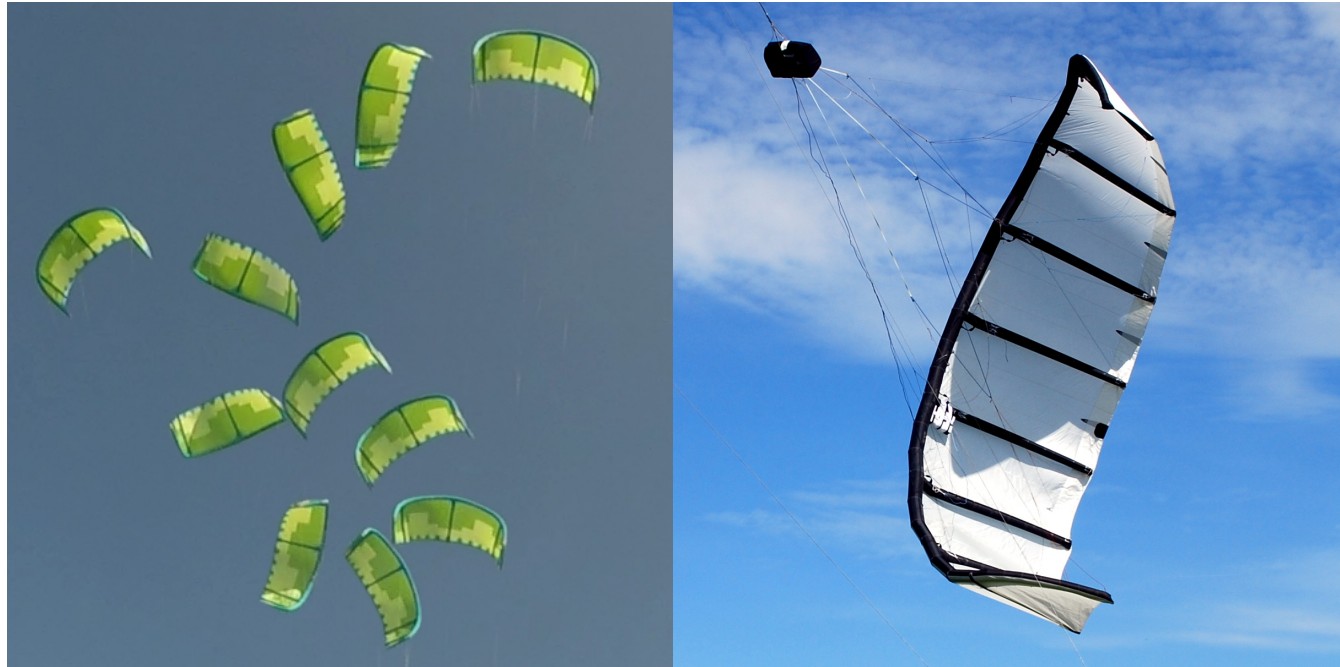

**Figure 1.** Time-lapse composite photo of a 6 m$^2$ power kite for sport applications performing a sharp turn to the left and subsequent loop manoeuvre, taken from the towing vehicle (left). Photo of a 25 m$^2$ power kite for airborne wind energy applications performing turn to the right during a mast-based launch manoeuvre, reported by Oehler and Schmehl (2019) (right). Both photos illustrate the spanwise twist deformation exhibited by LEI kites. The extreme deformation in the right photo does not occur in regular flight operations but resulted from mast-based launch attempts, shown here only to visualise the wing twist mechanism.

characteristic deformation is caused by pulling one steering line and releasing the other, with the two lines being attached to
the trailing edges of opposing wing tips. Breukels (2011) used an aero-structural model to simulate the deformation of an LEI kite during a turning manoeuvre. He concluded that this particular wing morphing is key to high manoeuvrability, proposing the following mechanism. Pulling on a wing tip while releasing the other locally increases the angle of attack of the pulled tip while decreasing the angle of the other tip. Also, pulling increases the effective vertical area of the respective wing half, while releasing decreases this area for the other half. Because of the twisting of the wing, the aerodynamic centres of the
tip regions are shifted relative to each other. The combination of these effects leads to a resulting aerodynamic side force and a yawing moment. The side force and the yawing moment are both important for initiating the turning manoeuvres. The described mechanism was confirmed by Bosch et al. (2014) using a fluid-structure interaction model of an LEI kite.



The effect of a steering input on the turn rate of a flexible membrane kite can be described by simple mechanistic models without considering a twist deformation or roll of the wing. A first "turn rate law" was formulated and experimentally validated for roll-controlled ram-air kites by Erhard and Strauch (2012, 2013b). Applying a simplified equilibrium of aerodynamic side force and centrifugal force during a turning manoeuvre, they found that the turn rate is linearly proportional to the product of steering line actuation and apparent wind speed at the kite. An extended formulation included an additional term accounting for the effect of gravity. Jehle (2012) and Jehle and Schmehl (2014) showed that this correlation approach also applies to LEI kites. Ruppert (2012) used a moment equilibrium at the wing tips to express the proportionality factor in terms of geometric and aerodynamic coefficients of the kite. Fagiano et al. (2013c, 2014) used a similar mechanistic model to identify another version of the turn rate law describing the rate of change of the velocity angle of the kite as a function of the steering input. Nowadays, the turn rate law is underlying many control algorithms for automated flight operation of flexible membrane wings (Vermillion et al., 2021; Fagiano et al., 2022). Flight tests have revealed that the dynamic response of the wing to a steering input can be significantly delayed, and some of the algorithms (Baayen, 2012; Rontsis et al., 2015; Costello et al., 2015, 2018; Wood et al., 2015) account for this delay with an additional term.

To experimentally determine the proportionality factor in the turn rate law, two approaches have been pursued: tow tests and flight tests using a fixed ground station with a winch. Wind tunnel measurements are not feasible because the dynamic flight state during a turning manoeuvre can not be recreated in a static setup, such as used in de Wachter (2008) and Rementeria Zalduegui (2019) for measuring the aerodynamic performance of kites and because a test section can spatially not accommodate a kite flying a turning manoeuvre. However, for outdoor measurements, the varying wind environment presents a challenge to the reproducibility of the measurement data. This is particularly important when the data is intended for parameter identification in simulation models (de Groot et al., 2011). The pioneering work of Erhard and Strauch (2012, 2013a) for Skysails was based on operational data from large ram-air kites of up to 320 m$^2$ wing surface area that were used to pull ships. The data included the steering input and the apparent wind speed at the kite, measured with an anemometer in the suspended kite control unit, as well as the turn rate of the wing. Jehle (2012) and Jehle and Schmehl (2014) used data sets from LEI kites with 14 and 25 m$^2$ wing surface area that were operated automatically in pumping cycles to generate electricity. A suspended kite control unit was used to steer the kite (see Fig. 1) and, in contrast to the concept of Skysails, also to power and depower the kite by changing the wing's angle of attack. Next to the wing's steering input and turn rate, the apparent wind speed was measured with a Prandtl tube suspended in the bridle line system of the kite (van der Vlugt et al., 2013). Oehler et al. (2018) used similar kites but now measured also the angles of the apparent wind velocity vector, using two orthogonal flow vanes, next to the previously mentioned properties and the position and orientation of the kite over time. They performed two different types of tests: fully automatic pumping cycle operation and flight manoeuvres at lower altitudes manually controlled from the ground. Borobia et al. (2018) performed flight tests with a 13 m$^2$ LEI kite that was manually controlled via four lines by a pilot on the ground, listing recorded time histories of steering input and rotation of the wing in space. Borobia-Moreno et al. (2021) substituted the simple Prandtl tube used in the earlier tests with a five-hole probe to also measure the inflow angles. Turning behaviour was not measured. Rushdi et al. (2020b, a) used a 6 m$^2$ LEI kite in a tow test setup to harvest data for machine learning. The kite was controlled by a suspended control unit that was attached to the towing vehicle via a short tether segment. Castelino et al.





(2022) used a 12 m² LEI kite that was steering manually with a control bar from a fixed ground point, estimating the tether force using deep neural network models. Schelbergen and Schmehl (2024) investigated the swinging motion of a kite with a

suspended control unit, as illustrated in Fig. 1(right), while flying turning manoeuvres. The swinging motion in terms of pitch and roll was modelled and compared to flight test data. The turning performance was not assessed in this work.

Hummel (2017) and Hummel et al. (2019) systematically investigated and quantified the effect of symmetric actuation for powering and depowering flexible membrane kites. The research was part of the TETA (Test and Evaluation of Tethered Airfoils) project, in which a tow test setup for reproducible flight experiments with flexible membrane wings was developed

(Duotone Kiteboarding, 2019; Kite Magazin, 2019; The Kiteboarder, 2019). To generate a constant and uniform flow field, the tests were conducted on days with very low ambient wind speeds. The towing vehicle and the trailer were equipped with several sensors and actuators to generate control inputs, fly reproducible manoeuvres, and measure various parameters.

The objective of the present study is to expand on this work by measuring the turning behaviour of kites using automatic and reproducible flight manoeuvres under controlled environmental conditions. The presented material is based on the PhD

research of the first author (Elfert, 2021). The paper is organised as follows. In Sect. 2, the measurement concept is presented. In Sect. 3, the test setup is detailed, and in Sect. 4, the data acquisition is described. This is followed by a presentation and discussion of the experimental results in Sect. 5 and an elaboration of conclusions in Sect. 6.

## 2   Measurement concept

To simulate a constant and uniform wind environment, the tests were executed on days with very low ambient wind, towing the

trailer-mounted test bench with constant speed along a straight track. The setup is illustrated schematically in Fig. 2, showing the different subsystems, components, and information flows. The test bench is mounted on a trailer and pulled by a vehicle.

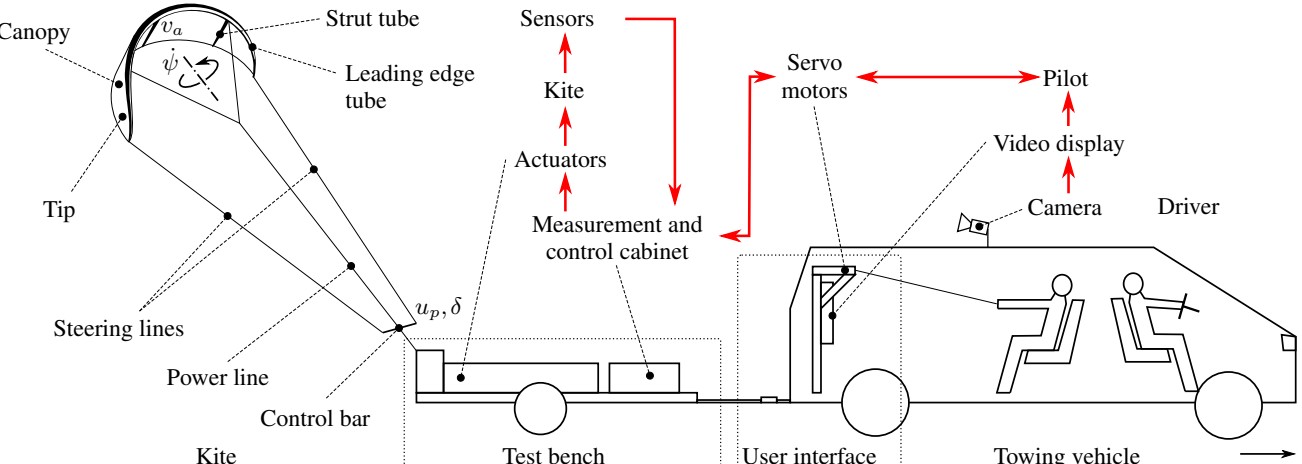

**Figure 2.** Towing test schematic, adapted from Hummel et al. (2019).





The kite is controlled by a pilot from within the vehicle, using visual feedback from a video camera. The manual steering input is converted into electronic signals transmitted to the test bench, which actuate the control bar. Sensors on the kite and part of the actuation mechanisms are fed back to the control cabinet. Servo motors provide force feedback to the pilot. This

specific decoupling of the pilot from the actuation of the control bar was chosen to allow for three different control modes: "fully manual", "semi-manual" and "fully automated". Because of the digitisation of the pilot's steering inputs, mixing them with outputs from the control algorithms is possible.

## 2.1   Measured properties

The analysis of the turning behaviour is based on the recorded time histories of the steering input $\delta$, the apparent wind speed $v_a$

and the resulting turn rate $\dot{\psi}$ of the wing. The power setting $u_p$ is considered an additional parameter in the analysis. Figure 3 details the definition of the power setting for symmetric actuation (left) and steering input for asymmetric actuation (right). The fully powered state is defined by $u_p = 0$ while the fully depowered state is defined by $u_p = 1$. When fully depowered, the

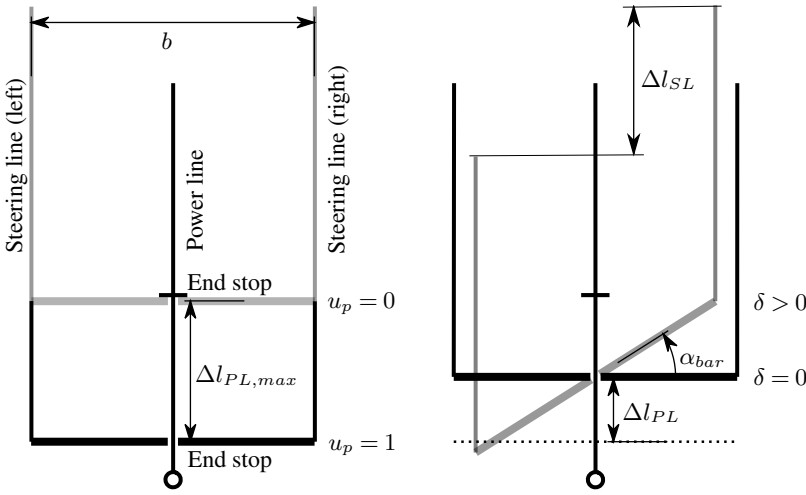

**Figure 3.** Definition of the power setting $u_p$ showing symmetric actuation in fully powered and depowered states (left) and steering input $\delta$ with associated asymmetric actuation at a half-way depowered kite (right).

length difference $\Delta l_{PL}$ between power and steering lines is at the maximum value $\Delta l_{PL,max}$. At any relative power setting $0 \leq u_p \leq 1$, the relative steering input is defined as

$$\delta = \sin \alpha_{bar}. \tag{1}$$

causing a length difference $\Delta l_{SL}$ of the two steering lines. Accordingly, $\delta$ can vary between -1 and 1.

Figure 4 illustrates the kinematic properties of a kite $K$ performing crosswind flight manoeuvres while being towed by a test bench $B$ along a straight track. The kite and the test bench velocities relative to a fixed point on the ground are denoted as



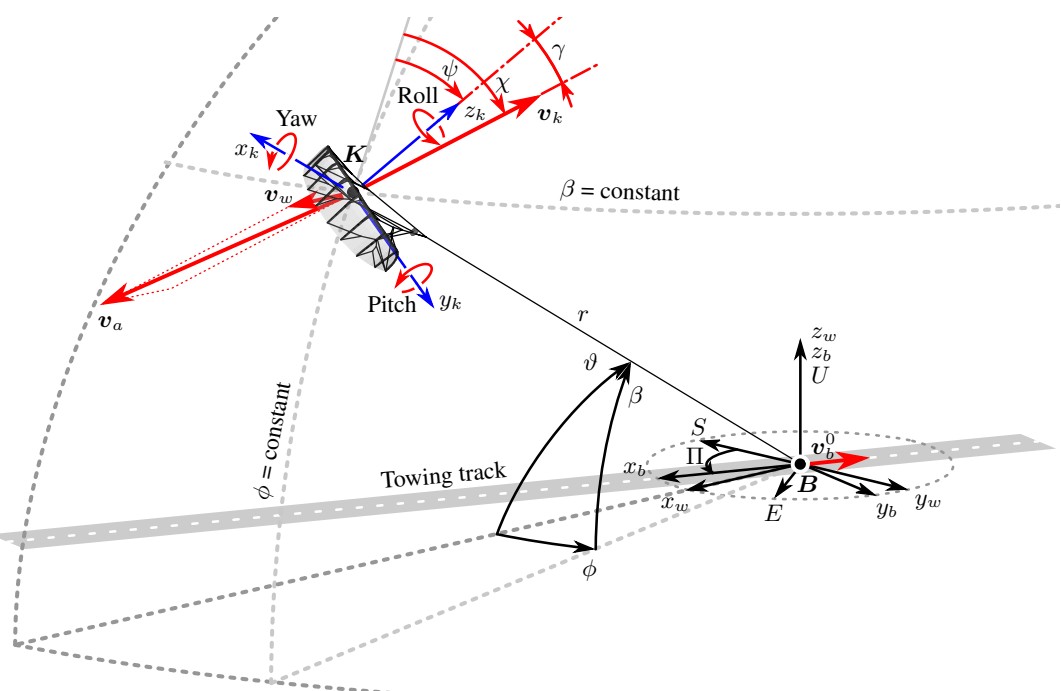

**Figure 4.** Kinematic properties of the kite $K$ and the towing test bench $B$ for a vanishing ambient wind speed $v_w^0$.

$v_k^0$ and $v_b^0$, respectively. The flight velocity of the kite relative to the moving test bench is thus given by

$$v_k = v_k^0 - v_b^0. \tag{2}$$

The wind velocity relative to the moving test bench is given by

$$v_w = v_w^0 - v_b^0, \tag{3}$$

where $v_w^0$ is the ambient wind velocity. Because the tow tests aim to simulate specific ambient wind conditions, the relative wind velocity $v_w$ is also denoted as simulated wind velocity. The corresponding relative wind reference frame $(x_w, y_w, z_w)$ is moving with the test bench, has its $z_w$-axis pointing vertically upwards, and its $x_w$-axis pointing in the direction of the wind velocity relative to the moving test bench. Also included in Fig. 4 is the south-east-up (SEU) Earth-fixed reference frame, and the body-fixed reference frame of the test bench is shown as $(x_b, y_b, z_b)$, with its $x_b$-axis aligned with the towing velocity $v_b^0$. The rotation of the $x_b$-axis from the south-direction is described by the angle $\Pi$. Equation (3) indicates that at low ambient wind velocity $v_w^0$, the simulated wind velocity $v_w$ can be controlled well by the towing velocity $v_b^0$.

The apparent wind velocity $v_a$ experienced by the kite, also denoted as inflow velocity, can be defined either in the moving or in the Earth-fixed reference frame as

$$v_a = v_w - v_k = v_w^0 - v_k^0. \tag{4}$$



The practical challenge in measuring $\boldsymbol{v}_a$ is not only the attachment of the flow sensor to the deforming kite system while maintaining a well-defined orientation but also the placement of the sensor in such a way that the flow is not disturbed by the

presence of the kite or the sensor itself (Oehler and Schmehl, 2019). The position of the kite in the wind reference frame is defined in terms of spherical coordinates $(r, \phi, \beta)$, where $r$ is the radial distance, $\phi$ is the azimuth angle, and $\beta$ is the elevation angle.

Figure 4 also includes the body-fixed reference frame $(x_k, y_k, z_k)$ of the kite and the associated rotations, denoted as yaw, pitch and roll. With this particular choice of $x_k$-, $y_k$- and $z_k$-axes, the reference frame coincides with the body-fixed reference

frame of the test bench $(x_b, y_b, z_b)$ when the kite is in its launch position at $\phi = 0$ and $\vartheta = 0$, sitting on its trailing edge, with the longitudinal axis of the kite pointing upwards. This enables an easy check-up of the calculated orientation angles before starting a measurement. Assuming a fully tensioned tether and a flight motion of the kite on the spherical surface with radius $r$, the orientation of the kite in a local tangential plane can be described by the heading angle $\psi$, while the orientation of the kite velocity can be described by the course angle $\chi$, which is coupled to the time derivatives of the azimuth and elevation angle by

the kinematic relation

$$\chi = \arctan\left(\frac{\cos\beta\,\dot{\phi}}{\dot{\beta}}\right). \tag{5}$$

This concept of describing the orientation of the kite and its velocity on a spherical surface originates from the field of navigation and has been applied in prior work on flight control (Jehle and Schmehl, 2014; Fagiano et al., 2014; Fechner and Schmehl, 2018). Both angles are measured clockwise from the upward (zenith) direction, ranging from $-180°$ to $+180°$. Within this

framework, the yaw rate of the kite is identical to the rate of change $\dot{\psi}$ of the heading angle, which is also denoted as the turn rate. The yaw angle and the velocity angle only differ if the kite is not aligned with the direction of flight. The drift angle $\gamma$ describes the difference between the velocity and orientation of the kite

$$\chi = \psi + \gamma. \tag{6}$$

The direction of the apparent wind velocity in the body-fixed reference frame of the kite is described by the angle of attack

$\alpha$ and the sideslip angle $\beta_s$. The angle of attack of the wing is measured in the $x_k z_k$-plane of the kite, while the side slip angle is measured in the $y_k z_k$-plane.

## 2.2 Turn rate law

The basic structure of the turn rate law can be derived from a simple mechanistic model of the kite flying a turn. This scenario is illustrated in Fig. 5 for a kite tethered to a fixed point on the ground. The model can be used equally for the towing test, but

then using the relative kinematics introduced in Sect. 2.1. For moderate steering actuation, it can be assumed that the angle of attack of the wing tip is linearly dependent on the steering input $\delta$. As a consequence, also the aerodynamic side force depends linearly on the steering input, which can be expressed as

$$F_{a,s} = C v_a^2 \delta,, \tag{7}$$



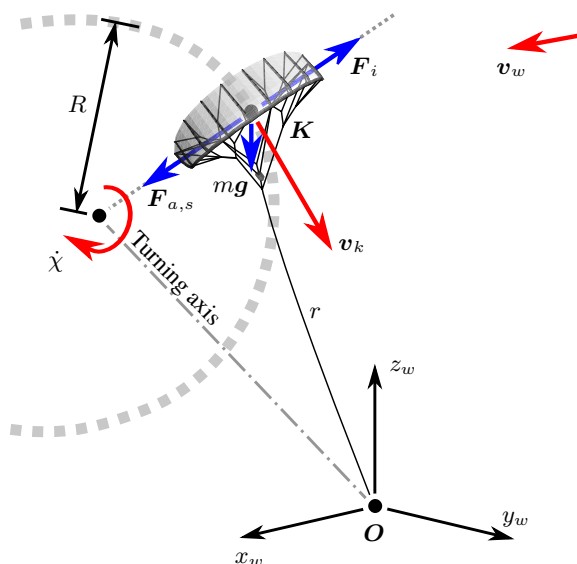

**Figure 5.** Kite flying a left turn with turning radius $R$ tethered to a fixed point $\boldsymbol{O}$ on the ground.

.

where all constant problem parameters are combined into a single constant $C$. How exactly the side force $F_{a,s}$ is generated, whether by roll or by asymmetric deformation of the wing, is not important for this analysis. Next, we can conclude from Fig. 5 that the rate of change $\dot{\chi}$ of the course angle and the kite velocity $v_k$ are coupled by the kinematic relation

$$v_k = \dot{\chi} R, \tag{8}$$

where $R$ denotes the turning radius. For steady flight on a trajectory with a constant curvature, the force equilibrium in the $y_k$-direction of the body-fixed reference frame of the kite can be formulated as

$$F_{a,s} + m\boldsymbol{g} \cdot \boldsymbol{e}_{y,k} = mR\dot{\chi}^2, \tag{9}$$

where $m$ represents the mass of the kite, $\boldsymbol{g}$ the gravitational acceleration, $\boldsymbol{e}_{y,k}$ the unit vector in $y_k$-direction and the term on the right-hand side the centrifugal force $F_i$. Inserting Equations (7) and (8) in Equation (9) leads to the course rate law

$$\dot{\chi} = g_k \frac{v_a^2}{v_k}\delta + \frac{\boldsymbol{g} \cdot \boldsymbol{e}_{y,k}}{v_k}, \tag{10}$$

where $g_k = C/m$ is a non-dimensional constant denoted as steering gain, depending on the geometry of the kite but also on various variable influencing parameters such as the line length, the position of the kite in the wind window, the inflow velocity and the power setting or the force in the steering lines. Oehler et al. (2018) showed on the basis of experimental data that the drift angle $\gamma$ is generally constant except for short periods at the start and end of a turning manoeuvre. From Equation (5), it can thus be concluded that the time derivatives of the course and heading angles are largely identical. This leads to the following



formulation of the turn rate law

$$\dot{\psi} = g_k \frac{v_a^2}{v_k} \delta + \frac{\boldsymbol{g} \cdot \boldsymbol{e}_{y,k}}{v_k}. \tag{11}$$

Assuming a fast-flying kite with $v_k \gg v_w$, the apparent wind speed can be approximated well by the flight speed of the kite, which leads to the basic, most frequently used formulation of the turn rate law

$$\dot{\psi} = g_k v_a \delta + \frac{\boldsymbol{g} \cdot \boldsymbol{e}_{y,k}}{v_a}. \tag{12}$$

Experimental studies have shown that the flight-dynamic response of the kite is delayed from the steering input, which is a dynamic effect that is not covered by the steady-state derivation above. The dead time is also not constant but depends on various influencing parameters. For this reason, we use the following expanded formulation of the turn rate law in the present study

$$\dot{\psi}(t) = g_k(t) v_a(t) \delta(t - d(t)), \tag{13}$$

where $d$ is the additional dead time. In this formulation, the influence of the gravitational force on the rotation rate is neglected, as proposed by Erhard and Strauch (2013b) for kites in crosswind flight where the gravitational forces are of minor importance compared to the dominant aerodynamic forces.

The steering gain $g_k$ and dead time $d$ represent the a priori unknown kite-specific parameters that depend on the current flight state of the kite described by the line length, the position in the wind window, the inflow velocity and the power setting or the force in the steering lines.

## 2.3 Measurement procedure

The tow tests were conducted on the asphalted shoulders of a straight, 1.5 km long runway section. The kite is positioned behind the test bench with taut lines for launching, sitting on its trailing edge. Once the test rig accelerates, the kite lifts off and is manually steered towards the zenith. The desired simulated wind velocity is set via the cruise control of the towing vehicle and controlled using data from a weather station mounted on the vehicle roof and the test bench's global navigation satellite system (GNSS) sensor. When reaching the target speed, the pilot uses a foot switch to activate the flight controller to perform reproducible flight manoeuvres. The activated controller allows fully- or semi-automatic flight depending on the specific settings. Once the end of the runway is reached, the flight manoeuvres are stopped, and the vehicle performs a U-turn while the kite is controlled manually. The same procedure is then repeated in the reverse tow direction and so forth, so measurement runs of any length can be conducted.

To accurately determine the steering gain and dead time, sufficient statistical averaging data must be collected for the entire range of wind power densities. This can be achieved by performing repeated figure-of-eight manoeuvres. Starting with $u_p = 0.4$, the power setting increases in discrete steps until reaching $u_p = 1$. For each step, a sufficient number of manoeuvres has to be executed. The shape of the figure-of-eight trajectories is adjusted so that the manoeuvres do not extend too far towards the lateral edge of the wind window. This region denotes the region of the spherical surface on which the kite can perform





quasi-steady flight manoeuvres (Schmehl et al., 2013). Because the reaction to steering inputs at low power settings may not be sufficient to counteract the gravitational force, it can happen that the kite can no longer be flown back into the wind window and drifts to the ground at the edge of the wind window. For high power settings, the kite can accelerate to high speeds and overpower the edge of the wind window, leading to stall and a subsequent crash of the kite. Both anomalies are prevented by using the robust flight control system presented in Sect. 4.1 for automated flight manoeuvres.

## 3 Test setup

This section further details the test setup, expanding on the work of Hummel (2017) and Hummel et al. (2019), adding an onboard sensor system to the kite. The purpose of this sensor is to acquire data about the flight state of the kite, including the position, velocity, orientation, and apparent wind velocity at the kite, which is a key parameter affecting the aerodynamic forces.

### 3.1 Test bench

Figure 6 shows a rendering of the test bench that is assembled on a single-axle car trailer. Two winches powered by servo

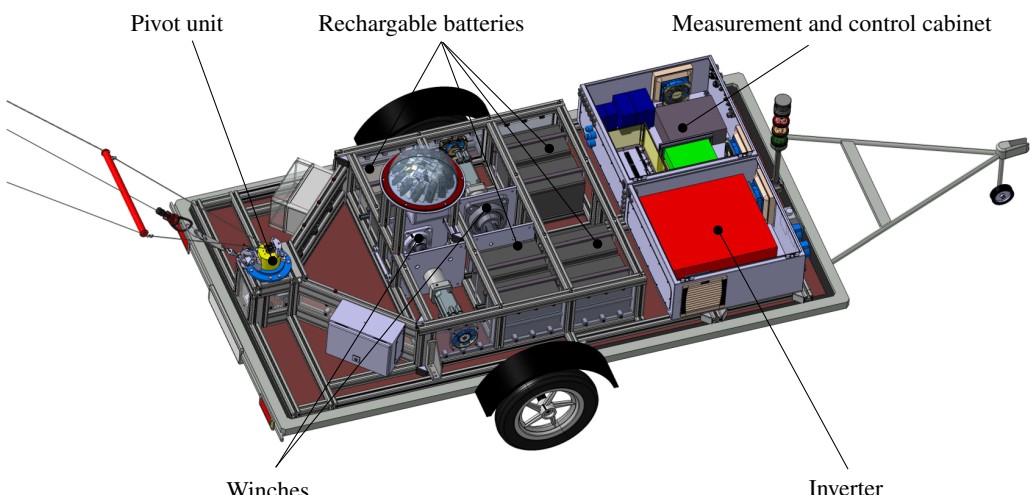

**Figure 6.** Test bench and its main components (Elfert, 2021).

motors are positioned in the centre of a frame made of aluminium profiles. Each motor is managed by a servo controller translating the commands received from the software via CAN and returning various status parameters of the motors. This also includes the rotational positions of the winches used to determine the length difference $\Delta l_{SL}$ of the two steering lines. Each 215 line is wound onto its winch and, from there, guided to the attachment points on the kite via two pulleys that are part of the pivot unit, illustrated in Fig. 7. This unit is also where the power line attaches, allowing the three lines to follow the movement





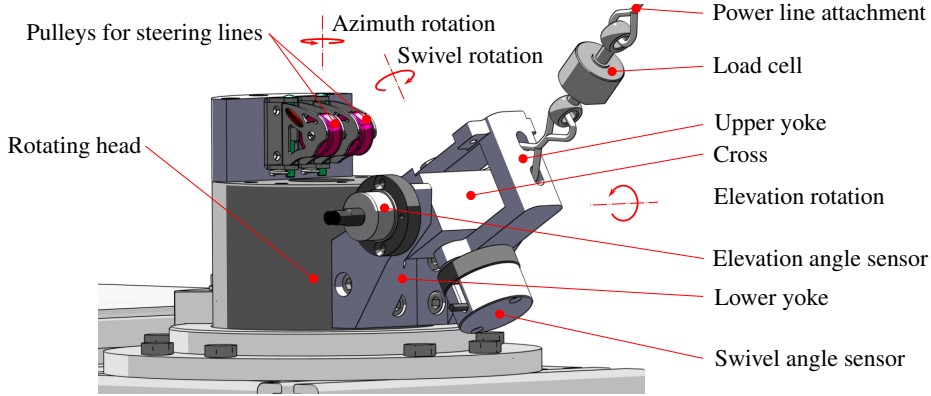

**Figure 7.** Detail of the passive pivot unit connecting to the power line via a universal joint (Elfert, 2021).

of the kite in the wind window, passively rotating within a range from $-135°$ to $+135°$ around the vertical axis. This azimuth rotation is measured by an angle sensor. The power line is connected to the rotating head via a universal joint that allows a rotation of $90°$ around the horizontal axis and a rotation of $±45°$ around the swivel axis, measured by additional angle sensors.

Assuming fully tensioned lines and concatenating the three successive azimuth, elevation and swivel rotations, the position of the wing in the wind window can be determined. The forces in the steering and power lines are measured with load cells. It is important to note that because of the additional swivel rotation, the elevation and azimuth angles recorded by the pivot unit do not correspond to the equally-denoted angular coordinates $\beta$ and $\phi$ used in Sects. 2.1 and 2.2.

A weather station is mounted on the roof of the towing vehicle to document the test conditions and set the desired simulated

wind velocity. The included cup anemometer and wind vane measure the magnitude and direction of the relative flow velocity generated by the towing. To avoid disturbances from the vehicle itself, the weather station is mounted on a tripod approximately 1 m above the roof. The weather station also measures the temperature, air pressure and humidity, and it is the only sensor system operating at a measurement frequency of only 1 Hz. All other data is acquired at a frequency of 50 Hz.

### 3.2 Sensor system

The sensor system consists of the kite sensor unit, shown in Fig. 8, and a base sensor unit mounted on the test bench. The data acquired by the kite sensor unit is transmitted wirelessly to the base sensor unit which, together with its own data, transmits this further to the test bench computer. The hardware components of the kite and base sensor units are largely identical and listed in Table 1.

Using GNSS sensors for position measurement of fast-flying kites on short lines is challenging, as frequently reported in the

literature (Fagiano et al., 2013a, b; Erhard and Strauch, 2013a). Within the frame of this project, it was considered that at least the GNSS data of the base sensor unit would be of great added value for the later data analysis. For example, it is helpful for a plausibility check to know what kind of route was followed on the airfield or at which point on the airfield certain measurement data were recorded. Since the kite sensor unit and the base sensor unit were to be built identically anyway to reduce complexity





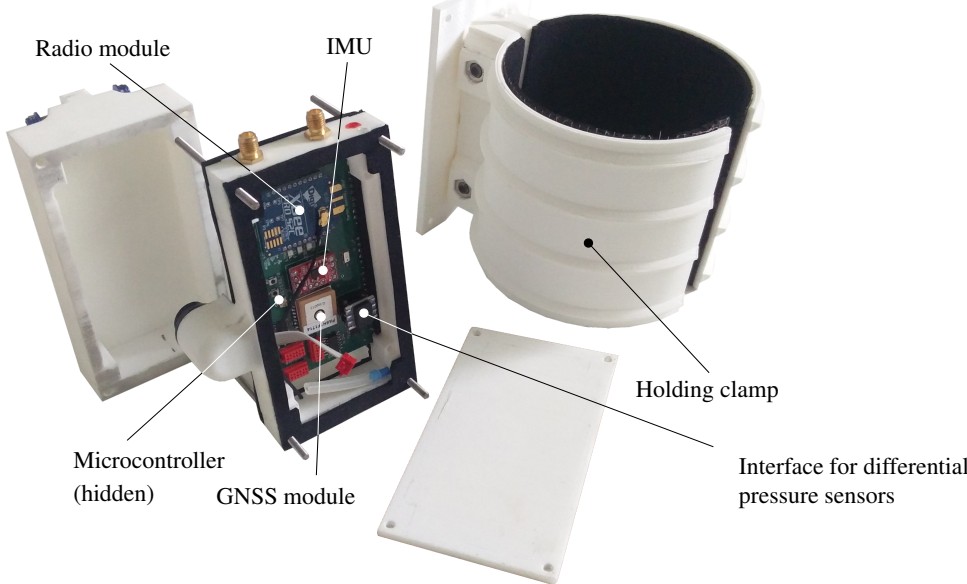

**Figure 8.** Components of the kite sensor unit (Elfert, 2021). The Teensy 3.6 microcontroller is not visible, as it is installed on the lower board.

**Table 1.** Hardware components of the kite sensor unit.

| Component | Specification |
| --- | --- |
| Microcontroller | Teensy 3.6 (see PJRC) |
| Integrated IMU | MPU-9250 (see InvenSense) |
| Radio module | XBee X2C Pro (see Digi) |
| GNSS-receiver | MT3339 (see Mediatek) |
| Barometric pressure / altitude sensor | MPL3115A2 (see NXP) |

and no disadvantage was to be expected by integrating the GNSS into the kite sensor unit in any case, the GNSS module was
also mounted on the board of the kite sensor unit.

The IMU data were post-processed with a fusion algorithm developed by Seel and Ruppin (2017). The settings of this algorithm have a direct physical meaning, representing the half-life period of the corresponding correction in seconds. In contrast to most IMU sensor fusion algorithms, it is possible to set separate weights for gravity-based and magnetic field-based estimation. Thus, for example, the negative influence of a magnetic field disturbed by external influences on the orientation
estimation can be reduced. Another special feature is an adaptive weighting of the influence of the measurement data of the acceleration sensors, which depends on the dynamics.




An alternative fusion algorithm was developed by Freter et al. (2020) to improve the kite position estimate. This is because a high-quality position estimate is a pre-condition for a good yaw angle estimate. The algorithm fuses the high-frequency and delay-free IMU data with the measured line angles, which allows for compensation of the inertia and sag of the lines. At the

same time, accurate measurement data of the line angle sensors serve as a reference to counteract the drift behaviour of the IMU estimation. Since the IMU measures in the kite-fixed reference frame, while the line angle sensors measure in the test bench-fixed reference frame, a coordinate transformation must be performed for a meaningful fusion. The algorithm of Freter et al. (2020) already includes a transformation from the kite's body-fixed reference frame to an Earth-fixed SEU-reference frame (cf. Fig. 4). When assuming, for simplicity, that the test bench moves in a perfectly horizontal plane, the bench-fixed

reference frame differs from the SEU-reference frame only by a rotation with angle $\Pi$ around the vertical axis. This angle can be calculated, for example, from the GNSS or IMU data of the base sensor unit, as shown in Fig. 9.

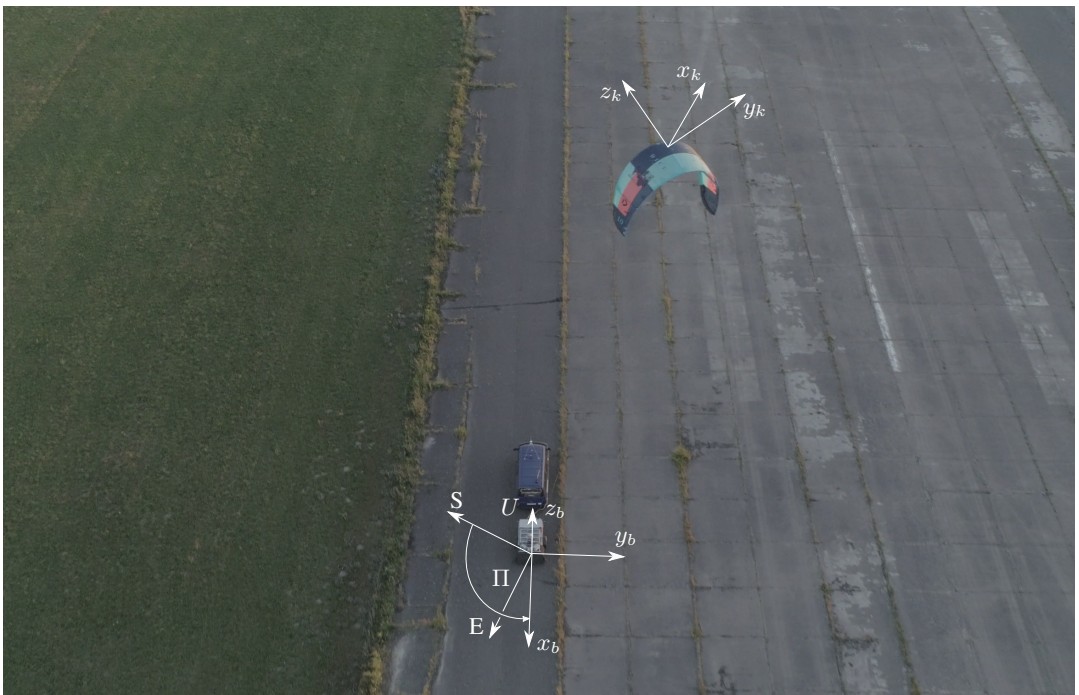

**Figure 9.** Kite-fixed coordinate system and test bench-fixed coordinate system (Elfert, 2021).

### 3.3   Relative flow sensor

The inflow velocity vector at the kite was measured with a custom-developed, low-cost five-hole Prandtl probe. A differential pressure flow meter is fairly accurate over a large measurement range compared to other techniques and accounts for varying

air density. Avoiding any moving parts, it can be designed to be reliable and robust. The developed probe is compact and lightweight, and its low inertia reduces the influence on the flight behaviour of the kite. The main disadvantage is the high cost





of commercial multi-hole Prandtl probes. To accurately measure the free stream flow velocity, the pressure openings of the probe must be positioned as far upstream of the kite's leading edge as possible, which means that the system is prone to be damaged in the event of a crash. For this reason, a low replacement cost for the probe was one of the key requirements for the development. Another challenge was that both the expected maximum values and the range of flow velocities are low in this application compared to the conventional application areas of such probes, such as aviation or racing. The Prandtl probe can be attached to the kite sensor unit and mounted together to the kite's central strut, as shown in Fig. 10.

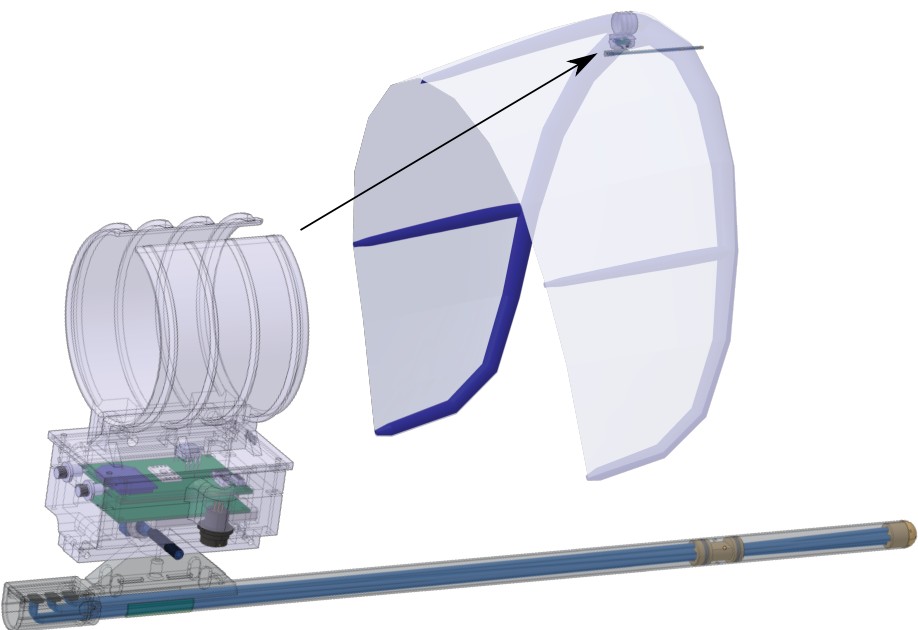

**Figure 10.** Onboard sensor system with Prandtl probe mounted on the kite (Elfert, 2021).

The orientation of the probe can be adjusted by exchanging the triangular adapter between the housing and the probe. This is adjusted in between flight tests so that the probe is aligned as much as possible with the inflow when in a non-manoeuvring flight. It should be mentioned that the angle of attack of the kite cannot be directly deduced from the measured angle of attack of the probe. For this, at least the probe's orientation with respect to a defined reference orientation on the kite must be known. In the case of fixed wings, this reference is typically the chord. However, kites deform strongly due to the actuation of the bridle line system and under the influence of aerodynamic loads, so a constant chord can not be assumed and thus at most a local angle of attack can be measured. Because the sensor system is mounted to the central strut tube, only the orientation of the probe or the inflow vector with respect to this part of the kite can be determined with certainty.

All necessary calculations to determine the magnitude and orientation of the inflow vector from the raw data of the differential pressure sensors are performed in the micro-controller of the kite sensor unit. Three differential pressures are calculated





from the raw data of the probe: The differential pressure of the centre hole to the holes for static pressure measurement and the differential pressure of each of the two opposite holes for the angle of attack and the sideslip angle. Using these, two non-dimensional coefficients are calculated, which are unique for a given orientation of the inflow vector and independent of the vector magnitude. To determine the angle of attack and the sideslip angle from these coefficients, we created a lookup table from wind tunnel experiments in which we adjusted the probe's orientation relative to the flow.

The measured centre pressure difference starts to deviate from the dynamic pressure at angles of incidence of more than $10°$. This happens because the holes for static pressure measurement are more and more in the incident flow, and the centre pressure hole is slowly rotated out of the incident flow. At $45°$, this phenomenon becomes so strong that the differential pressures become negative. To calculate the actual dynamic pressure, the measured centre pressure difference is therefore scaled depending on the calculated flow angles. Scaling is also used to correct the pressure loss in the tubes, which depends on the inflow velocity and the tube length. The correction polynomial used for this is determined from the measurement data measured in the wind tunnel at different incident flow velocities for a flow-aligned probe.

## 4 Data acquisition and analysis

In the following, the flight control algorithm developed specifically for this project and the post-processing of the raw data are presented in more detail.

### 4.1 Control algorithm for robust automated flight manoeuvres

The purpose of the flight controller is to ensure reproducible figure-of-eight manoeuvres in a wind environment with disturbances, such as turbulence and gusts. Our aim was to develop an algorithm that requires no priori parameter knowledge of the kite. It should further be possible to intuitively adjust the shape of the figures via the controller parameters. Fagiano et al. (2014) proposed an algorithm that fulfils these requirements, relying only on line angle sensor data. In this case, the controlled variable is the velocity angle of the kite, defined as the angle between the kite velocity vector and a fixed reference vector.

The selected algorithm generates the figure-of-eight path by alternatingly switching between two different target points $\boldsymbol{P}_+$ and $\boldsymbol{P}_-$, as illustrated in Fig. 11 and presented in more detail in Lange (2018). We decided to fly the kite upwards during the turning manoeuvres because the probability of crashes during these "up loops" is lower than during the alternative "down loops". The target points are switched before they are reached. To enforce that the kite flies "up loops" after switching the target point, it is forced to turn towards the zenith. The course angle $\chi_p$ required to reach the active target point $\boldsymbol{P}$ is determined from the current position of the kite using nautical course angle calculation. In contrast to Fagiano's original algorithm, the course is not calculated using the rhumb lines but the orthodromes. Initial field tests showed that this modification can lead to smoother and more robust flight manoeuvres. Using the orthodromes, the sphericicity of the kite's surface of motion is taken into account, which Fagiano did not do.

The shortest distance on the spherical surface between the kite $\boldsymbol{K}$ and the target point $\boldsymbol{P}$ is given by the great circle arc, denoted as orthodrome and indicated by the red-dotted line in Fig. 11. For a unit sphere, the length of the orthodrome is





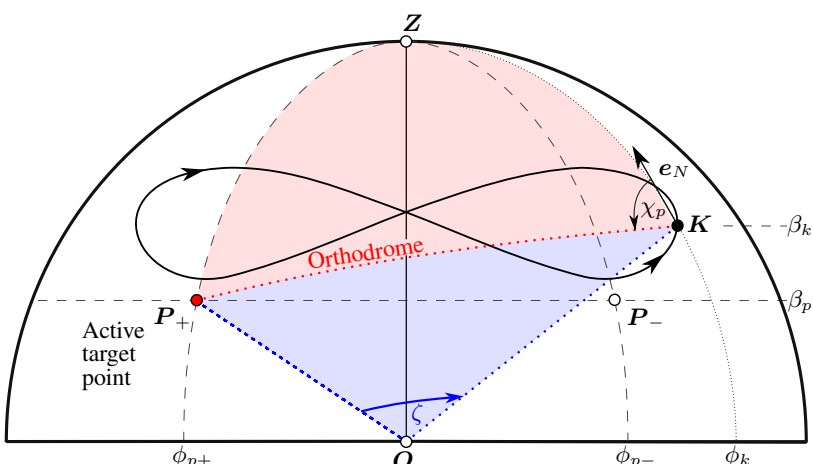

**Figure 11.** Calculation of the required course angle using the orthodrome, adapted from Fagiano et al. (2014); Lange (2018).

numerically equal to the angle $\zeta$ spanned by the radial lines to the two points on the surface. In Fig. 11, the respective great circle sector is shaded in blue. The angle $\zeta$ can be derived from the law of cosines for the red-shaded spherical triangle spanned by points $K$, $P$ and $Z$ on the unit sphere with the corresponding opposite sides $90° - \beta_p$, $90° - \beta_k$ and $\zeta$. The formulation for the angle $\phi_p - \phi_k$ with opposite side $\zeta$ leads to (Gellert et al., 1975)

$$\cos \zeta = \sin \beta_k \sin \beta_p + \cos \beta_k \cos \beta_p \cos (\phi_p - \phi_k). \tag{14}$$

Similarly, the formulation for the initial course angle $\chi_p$ with opposite triangle side $90° - \beta_p$ leads to

$$\cos \chi_p = \frac{\sin \beta_p - \sin \beta_k \cos \zeta}{\cos \beta_k \sin \zeta}. \tag{15}$$

Using this initial value, the set point is defined as Fagiano et al. (2014)

$$\gamma_p = \begin{cases} \chi_p, & \text{for} \quad \phi_k \leq \phi_{p\pm}, \\ -\chi_p, & \text{for} \quad \phi_k > \phi_{p\pm}. \end{cases} \tag{16}$$

Because the switching between target points leads to abrupt changes of the reference variable, a low-pass filter is connected downstream, which softens these jumps of the target heading angle before it is transferred to the inner loop of the controller. In this way, a steady evolution of the reference variable is achieved, and the rate of change is adapted to the inherent dynamics of the kite. A block diagram of the cascaded velocity angle controller is shown in Fig. 12.

In the inner loop, the feedback of the control deviation takes place via a simple P-controller. A manipulated variable limitation occurs in the downstream saturation block following the P-controller. The current velocity angle of the kite required for calculating the control deviation is calculated from Equation (5). The velocity angle is thus defined in a range of $\chi \in [-180°, 180°]$. This can cause erroneous calculations in the form of jumps during the velocity angle. In certain flight situations, for example, when the kite is held steady in the zenith, the kite can fly backwards in the wind direction for a short time.





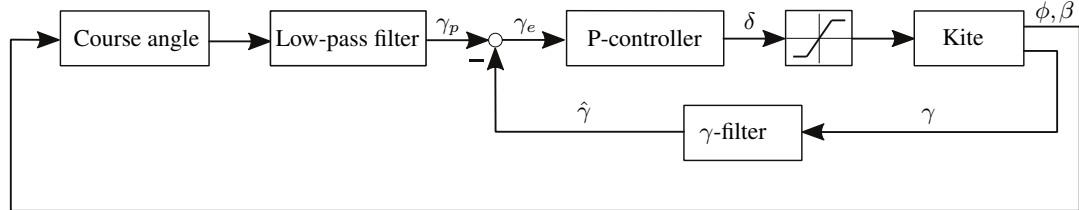

**Figure 12.** Block diagram of the velocity angle controller, adapted from Fagiano et al. (2014).

In this case, the value of the velocity angle would jump from $0°$ to $180°$ or $-180°$. The noise of the line angle sensors can lead to similar misinterpretations. To prevent such physically impossible jumps in the trajectory of the kite, an additional filter is used. The size and robustness of the eighth-shaped trajectories generated with this algorithm can be influenced by shifting the target points, the choice of the gain factor, and the setting of the low-pass filter.

### 4.2 Measurement data analysis

The steering gain and dead time were determined using a rearranged version of Equation (13)

$$\frac{\dot{\psi}(t)}{v_a(t)} = g_k \delta(t - d). \tag{17}$$

The time-dependent left-hand side of this equation differs from the time-dependent steering input $\delta(t)$ on the right-hand side only by a scaling factor $g_k$ and a time delay $d$, which we both assume to be functions of the power setting of the kite. Figure 13 illustrates two typical signals recorded during the towing on the straight track between two U-turns. In this particular example,

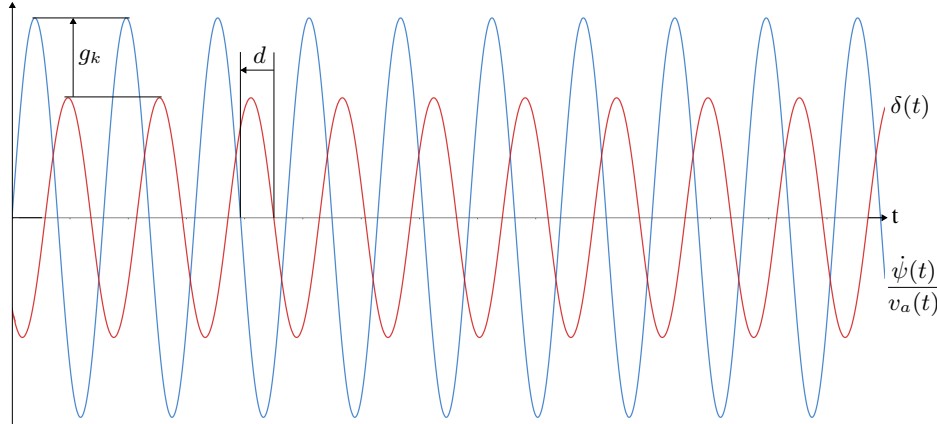

**Figure 13.** Typical time-dependent contributions to the left- and right-hand side of Equation (17) between two U-turns of the towing vehicle (Elfert, 2021).



ten manoeuvres were performed between the two turning points. The steering gain and dead time can be determined from the two time-dependent signals by numerical optimisation, fitting the right-hand side of Equation (17) for the considered
measurement period as closely as possible to the left-hand side. For this purpose, we used a simplex method with maximum likelihood estimation (MLE).

The data analysis for a specific power setting is illustrated schematically in Fig. 14. Especially at low power settings, the kite

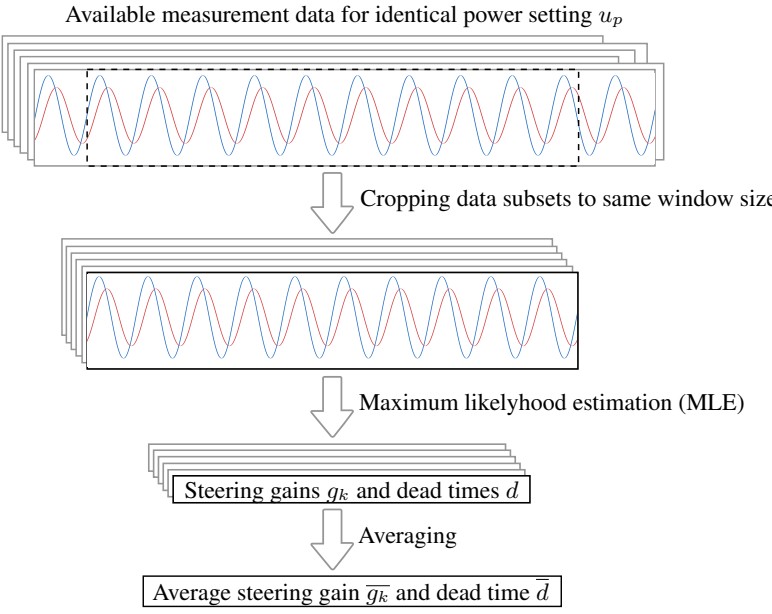

**Figure 14.** Data analysis for flight manoeuvres with identical power setting, adapted from Elfert (2021).

needed some time to settle in and converge to uniform flight manoeuvres. The data available from different measurements were cropped off at the start to remove the effects of these transient start-up phases. To minimise the effect of random disturbances,
the data subsets were kept as large as possible, cropping them at the end to the same size. The MLE was then applied to each subset to determine the respective steering gain and dead time. Of these, the mean values were finally calculated as representative of the specific power setting of the kite.

## 5   Experimental results

The measurements were conducted on 15 September 2020 at the former airfield in Pütnitz, Germany, using a Vegas 2015
kite from Duotone Kiteboarding (formerly North Kiteboarding) with a flattened wing surface area of approximately 10 m$^2$. Figure 15 shows the kite with a mounted Prandtl tube. The design geometry is summarized in Table 2. During the measurement runs the ambient wind speed $v_w^0$ was generally up to 3 m s$^{-1}$, while some rare wind gusts would occasionally increase this to 5.8 m s$^{-1}$. For all measurements, the simulated wind speed $v_w$ was set by the towing vehicle's driving speed $v_b^0 = 11.3$ m s$^{-1}$





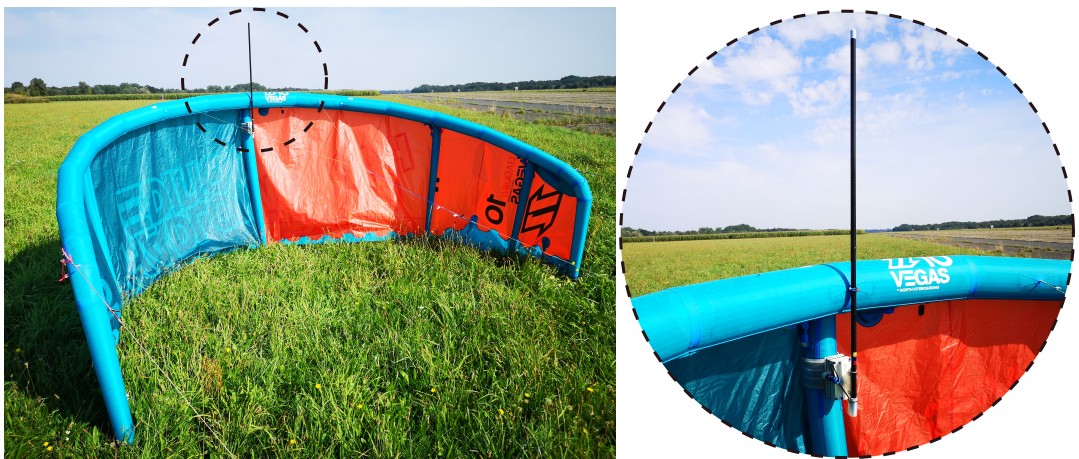

**Figure 15.** The Vegas 2015 kite with mounted Prandtl tube, adapted from Elfert (2021).

(22 kn). This specific speed was chosen because it is the median of the suitable wind speed range specified by the manufacturer.

It can thus be assumed that the aerodynamic loading and the respective flight behaviour are well within the range for which the kite was designed.

The maximum length difference between power and steering lines was $\Delta l_{PL,max} = 500$ mm. The power range for which the kites were designed is in fact more narrow, meaning that we were able to analyse also off-design actuation scenarios with very low power setting. In these cases, we observed that the steering lines were sagging so much that even the maximum

steering input would only straighten the lines without actually actuating the wing, leaving it unresponsive to the control input. At slightly higher power settings, it was possible to generate a small yaw movement with maximum steering input, but this was insufficient for controlled figure-of-eight flight manoeuvres.

For this reason, we first had to iteratively determine the lowest feasible power setting before conducting any measurement runs. Then, the target points and the controller gain factor had to be adjusted until it was possible to fly uniform and sufficiently

robust manoeuvres. For the kite used, a minimum power setting of $u_p = 0.4$ was determined.

**Table 2.** Design geometry of the tested Vegas 2015 kite (values taken from the largely similar 2018 model).

| Geometric property | Unit | Value, flattened | Value, projected |
|---|---|---|---|
| Surface area | m$^2$ | 9.7 | 5.73 |
| Aspect ratio | - | 5.2 | 2.48 |
| Span | m | 7.1 | 3.78 |



## 5.1 Measured steering gain and dead time

Following the procedure outlined in Sect. 2.3, the entire range of power settings of the kite was covered with a single measurement run by increasing the power setting by $\Delta u_p = 0.1$ at every second U-turn, starting at the lower limit $u_p = 0.4$ until reaching the fully powered state at $u_p = 1$. Consequently, every tow direction was measured once per power setting. As described in Sect. 4.2, a data window of fixed size was used for the analysis between two turning points, resulting in one window per power setting and tow direction. We found the measurement cycles 1000 to 3500 to span a suitable data range, excluding the transient start-up phases. Because the power setting affects the aerodynamic properties and by that also the flight speed of the kite (Hummel et al., 2019), the number of figure-of-eight manoeuvres per data window varied with the power setting.

Figure 16 shows the resulting average steering gain and dead time as functions of the power setting for the first measurement run. The expected increase in the kite's agility with the power setting is evident from the pronounced monotonic increase of the

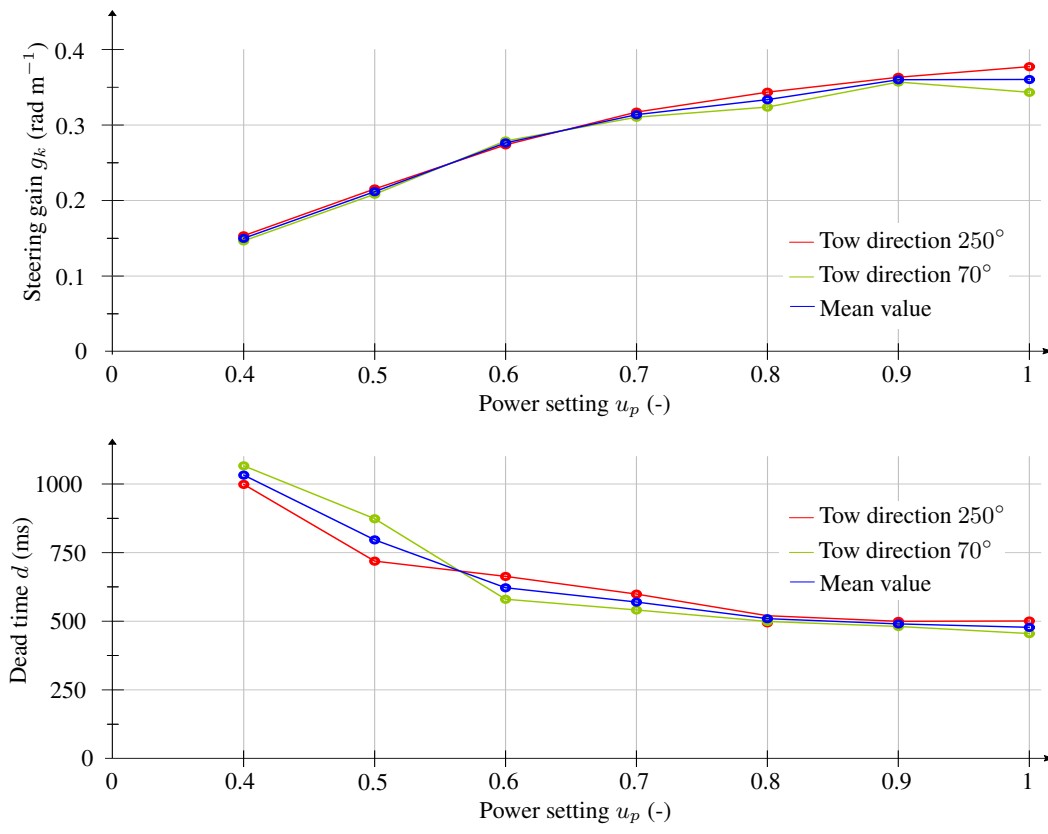

**Figure 16.** Steering gain $g_k$ and dead time $d$ as functions of the power setting $u_p$ for the kite Vegas 2015 with $10\ \mathrm{m}^2$ wing surface area. Adapted from Elfert (2021).

steering gain and decreased dead time. The higher the power setting, the faster and stronger the reaction of the kite to control




inputs. The main source of experimental uncertainty was estimating the yaw angle $\psi$ using the fusion algorithm described in Sect. 3.2. Additional uncertainty was introduced by the use of numerical optimisation, which is part of the MLE algorithm described in Sect. 4.2.

## 5.2 Sensitivity analysis

Several sensitivity analyses and plausibility checks were conducted to increase confidence in the experimental results. We first assessed the influence of the initial values on the optimisation result of the MLE, finding a significant effect only for strong deviations from realistic initial values. This observation is illustrated in Fig. 17, where the red line resulted from reasonable initial values $g_{k,0} = 0.3$ rad m$^{-1}$ and $d_0 = 600$ ms, and the blue line from extreme initial values $g_{k,0} = 10$ rad m$^{-1}$ and $d_0 = 1200$ ms. It is evident that the red line follows the noisy raw signal $\dot{\psi}/v_a$ closer in magnitude and phase shift.

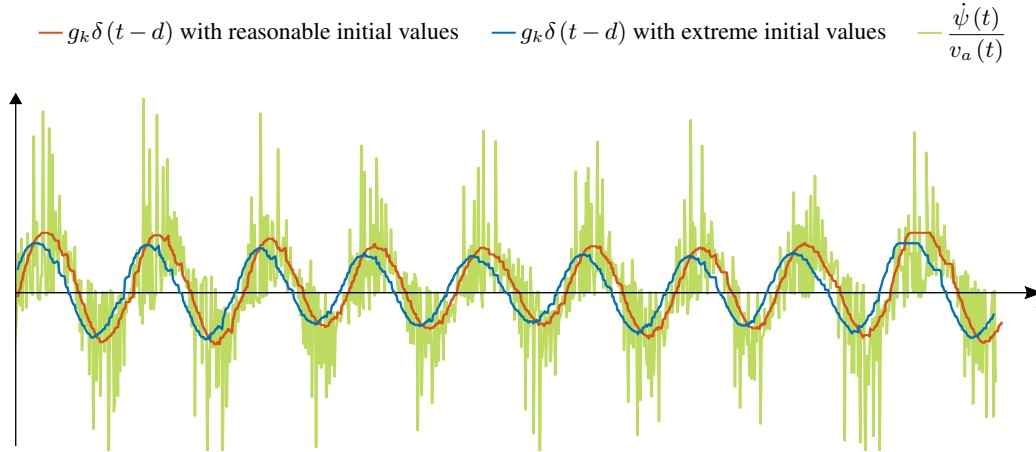

**Figure 17.** MLE optimisation with reasonable (red) and extreme (blue) initial values, adapted from Elfert (2021).

Figure 16 shows that the calculated turning characteristics vary slightly for the two tow directions, 250° and 70°. Suspecting that this was caused by variations in the inflow velocity due to ambient wind or thermals, we conducted a sensitivity analysis to assess the influence of such variations. As a first step, the plausibility of the inflow velocity measurements was investigated because the measurement campaign was in fact the first occasion that the custom-developed Prandtl probe was used in field tests. The inflow speed was checked by stopping the crosswind flight manoeuvres, i.e. setting $v_k$ in Eq. (4) to 0, and towing the kite in a static flight position relative to the vehicle. The inflow speed measured in this flight mode was roughly 1 to 2 m s$^{-1}$ below the airspeed measured by the weather station on the towing vehicle.

The angle of attack and sideslip angle were checked by investigating the data of the first measurement run. Both diagrams in Fig. 18 include the manoeuvre release signal and the power setting indicating when the kite was operated in automated figure-of-eight flight manoeuvres and at which power setting. Two manoeuvre releases are executed for each power setting step, corresponding to the two driving directions, 250° and 70°. As expected, the mean angle of attack decreases with increasing



**WIND
ENERGY
SCIENCE
DISCUSSIONS**

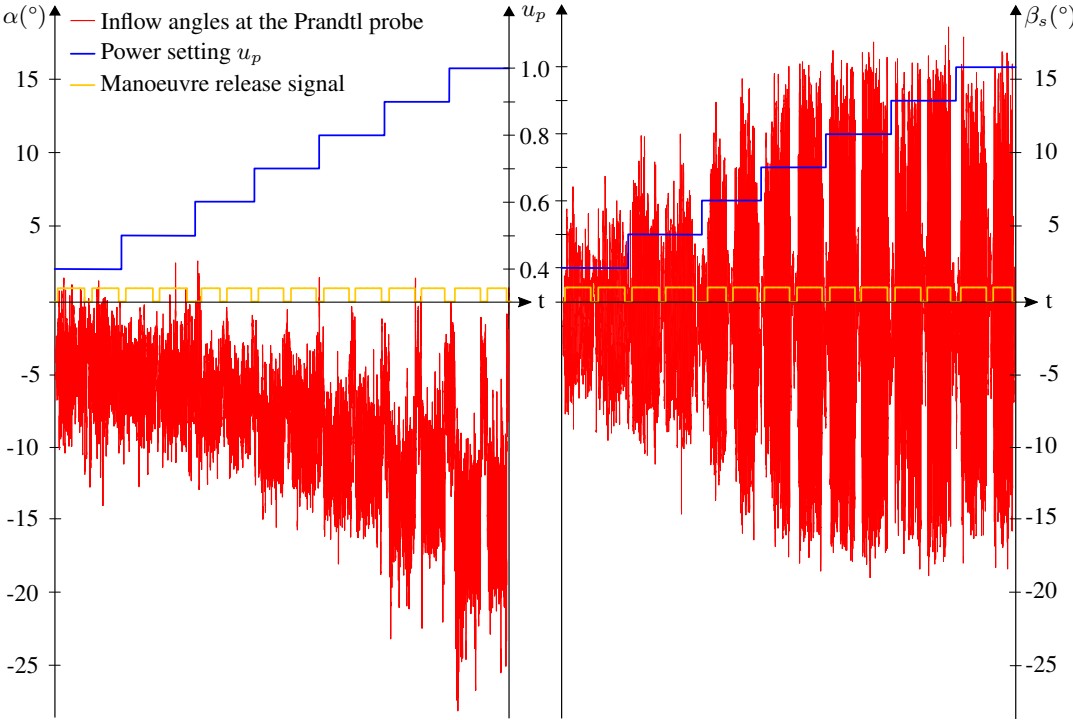

**Figure 18.** Inflow angle of attack $\alpha$ (left) and inflow sideslip angle $\beta_s$ (right) at the Prandtl probe. Adapted from Elfert (2021).

power setting[1]. It is also evident that the range of variation during crosswind flight increases with the power setting. This also holds for the sideslip angle, which, for power settings above 0.6 to 0.7, varies roughly between -15° and +15°. This range corresponds with what is known from previous studies on crosswind operations of kites (Ruppert, 2012; Fechner and Schmehl, 2018; Oehler and Schmehl, 2019).

Figure 19 compares the time histories of the sideslip angle and the yaw angle, showing a strong correlation and also an increasing range of variation of the yaw angle with the power setting. For a low power setting of 0.4, the yaw angle varies between -40° and +40°, which means the kite just sways left and right without completing any loop manoeuvres. For higher power settings, the range of variation increases but only occasionally reaches the extreme values -90° and +90°. Because a complete turning manoeuvre requires the yaw angle to vary by 180°, while the straight-path sections of a figure-of-eight even

expand this range by 10 to 30° in positive and negative directions, it is clear that even at higher power settings, the kite only sways left and right, rarely completing full loop manoeuvres. The main reason for this was the relatively high elevation angle (76 - 78°) during towing, which effectively depowers the kite and lowers its flight speed.

---

[1]The inflow angles $\alpha$ and $\beta_s$ describe the orientation of the inflow relative to the probe, while the corresponding attitude angles describe the probe's orientation relative to the inflow. For this reason, the inflow and attitude angles are defined with opposite sign.





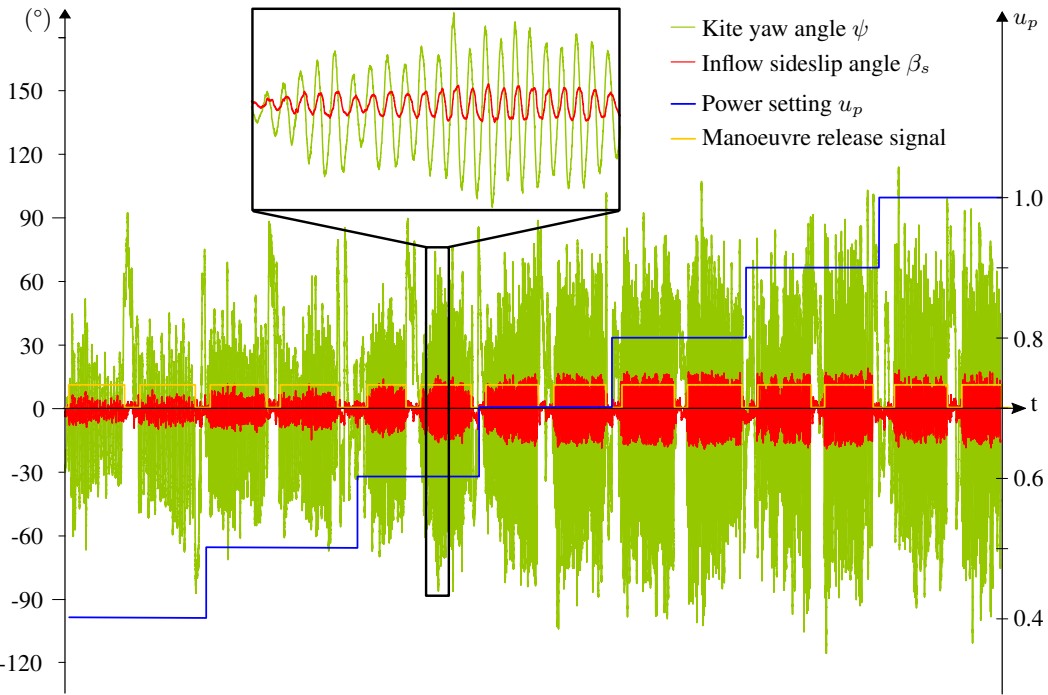

**Figure 19.** Inflow sideslip angle (red) at the Prandtl probe and yaw angle (green) of the kite calculated by the kite sensor unit. Adapted from Elfert (2021).

We also observed that contrary to expectations (see the discussion in Sect. 3.2), the kite-mounted GNSS module worked reliably also during highly dynamic flight manoeuvres. This can be seen in Fig. 20, where we compare the measured inflow velocity and the flight velocity determined by the GNSS module. This "GNSS velocity" is measured in the Earth-fixed reference frame. From Eq. (4) it is clear that the GNSS velocity $v_k^0$ and the apparent wind speed $v_a$ are identical for a vanishing ambient wind speed $v_w^0$. It is evident that the time history of the GNSS velocity closely follows the time history of the inflow velocity with only a small offset of around 2 to 3 m s$^{-1}$. The two velocities peak in the short breaks between the manoeuvre sequences. This increase is caused by the U-turns of the towing setup, when the crosswind manoeuvres are discontinued, and the kite is positioned at the lateral edge of the wind window performing 180° turns at the substantially larger radius and correspondingly higher speed than the towing vehicle. The diagram also shows that the range of velocity fluctuations increases with the power setting.

### 5.3 Detection and correction of faulty data

The Prandtl probe and the kite-mounted GNSS sensor provide velocity data from two entirely different measurement techniques. The data can thus be combined to detect and correct faulty data, further reducing the uncertainty of the individual measurements. Figure 21 compares the mean turning performance $g_k$ and mean dead time $d$ calculated using either the inflow

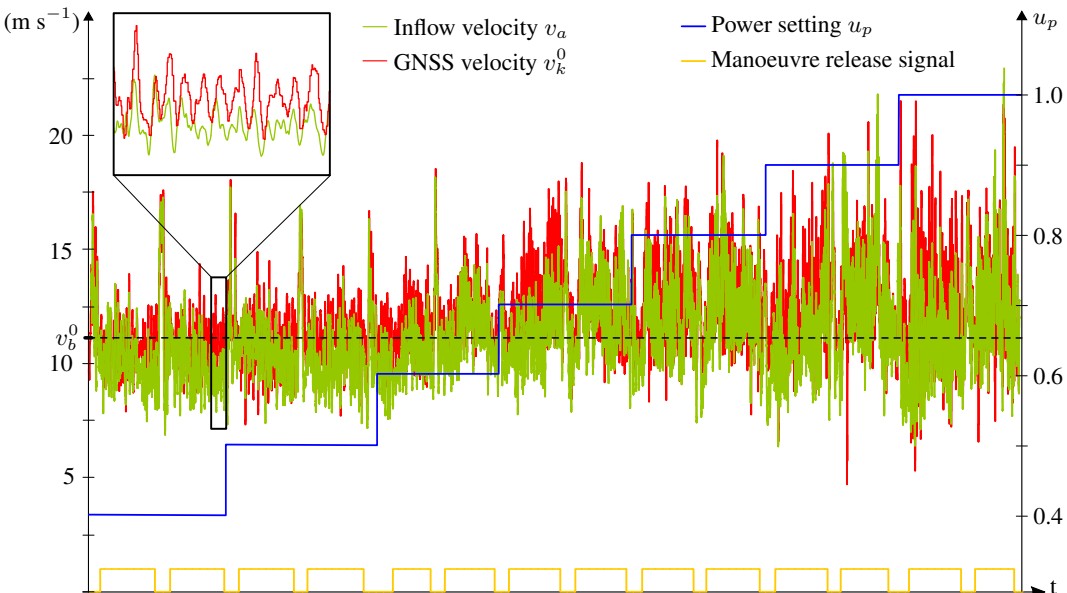

**Figure 20.** Inflow velocity measured at the Prandtl probe (green) and flight velocity (red) determined by the GNSS module. Tow speed set to $v_b^0 = 11.3$ m s$^-$1. Adapted from Elfert (2021).

velocity or the GNSS velocity for the MLE algorithm. The reported offset between the inflow and GNSS velocities leads to a steering gain offset $\Delta g_k$ ranging from 0.01 to 0.04, affects the mean values of the calculated turning performance in the form
of an offset $\Delta g_k$ ranging between 0.01 and 0.04. As expected, this offset does not significantly affect the calculated dead time.

The steering gains plotted in Fig. 16 for the two tow directions deviate most noticeably at the maximum power setting $u_p = 1$. To determine whether this deviation was caused by the ambient wind velocity, the measured flow velocities were investigated in more detail. From Fig. 22, it can be seen that the time histories of the simulated wind velocity and the inflow velocity do not differ much between the two opposing tow directions. The small average velocity difference of roughly 1 m s$^{-1}$ does not
explain the significant offset $\Delta g_k$ of 0.03 (8.5%) at the maximum power setting.

The actual cause for the offset of the steering gain can be traced back to the distinct peak of the inflow velocity in the last third of the second manoeuvre (tow direction 70°) illustrated in Fig. 22. An analysis of the position data showed that a cornfield about 3 m high was reached precisely at this point, which was the only landscape irregularity in the direct vicinity of the runway. Since high temperatures around 27° C and cloudless skies prevailed on the day of the measurement campaign, it is
reasonable to assume that the peak of the inflow value was caused by thermals above the cornfield. Because the simulated wind speed was measured by the weather station mounted on the roof of the vehicle, so, much closer to the ground, this ambient wind field perturbation at the height of the kite does not show in the time history of the simulated wind speed.

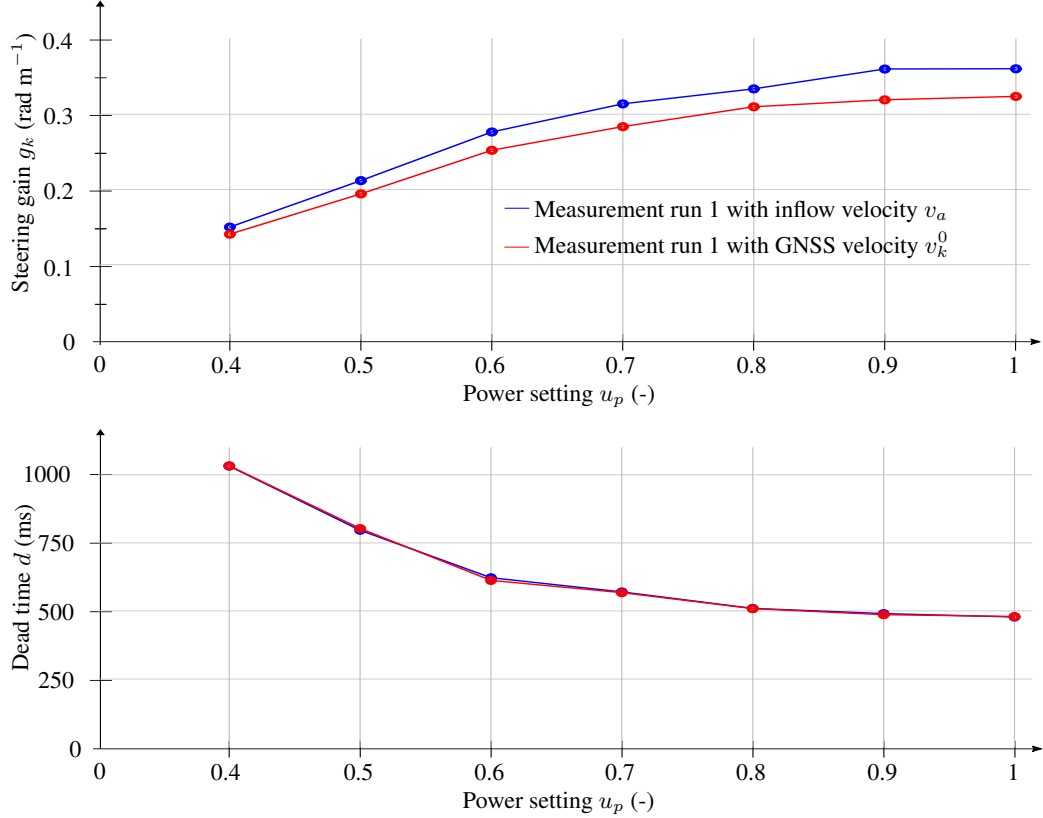

**Figure 21.** Comparison of the steering gain $g_k$ and dead time $d$ averaged over both tow directions as functions of the power setting $u_p$ for the Vegas 2015 with 10 m$^2$ when performing the MLE with the inflow velocity (blue) and GNSS velocity (red). Adapted from Elfert (2021).

A more detailed analysis of additional measurement channels revealed that the wind gust influenced the kite so strongly that the controller was thrown out of rhythm, not switching target points anymore (cf. Sect. 4.1). For high power settings, the controller gain factor was set to borderline aggressive and the sudden perturbation led to an overshoot of the controller so that the kite only oscillated around the right target point instead of alternating between the target points. Since the supervising pilot did not notice this anomaly in the video recording with the GoPro camera, the manoeuvre was not aborted, and the measurement data were accepted as valid and used for the postprocessing. The steering gains $g_k$ calculated with this data most likely turn out to be lower because the anomaly occurred close to the lateral edge of the wind window, where the inertia of the kite has a stronger effect on the steering behaviour. The examination of all manoeuvre runs of the first measurement run showed that the described faulty behaviour of the controller also occurred at the power settings $u_p = 0.9$ and $u_p = 0.8$. Just as with $u_p = 1$, the error occurred exclusively in the 70° direction of travel and in the area of the cornfield. In contrast to $u_p = 1$ and $u_p = 0.8$, however, the error in $u_p = 0.9$ does not have a noticeable effect on the calculated steering gain $g_k$. This can be

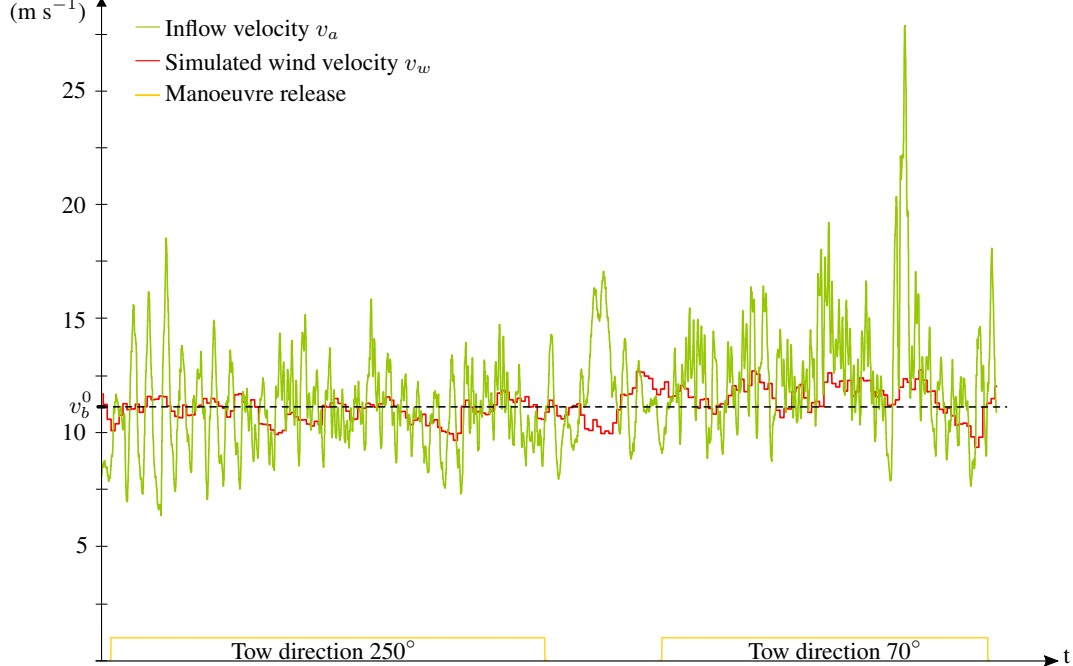

**Figure 22.** Time histories of the simulated wind velocity measured by the weather station as well as the inflow velocity measured by the Prandtl probe during the last two manoeuvres in Fig. 20 with power setting $u_p = 1$. Tow speed set to $v_b^0 = 11.3$ m s$^-$1. Adapted from Elfert (2021).

explained by the fact that in this manoeuvre run the error did not occur directly at the edge of the cornfield but somewhat later
so that the associated measurement data are no longer in the manoeuvre period selected for the evaluation.

To verify this hypothesis, the six manoeuvre runs of the three power settings concerned were evaluated separately using the measurement data of longer periods (manoeuvre cycle 1000 to 4500). Using this extended data, we could reproduce the described anomaly and the expected influence on the calculated steering gain $g_k$ for the power setting $u_p = 0.9$. Revisiting Fig. 16, it can thus be concluded that the steering gains derived from the measurement data of the 250°-tow direction are of
higher quality.

### 5.4   Second measurement run

Following the first measurement run a second run was attempted. Unfortunately, an incorrect initialisation of the control algorithm led to an early crash of the kite, damaging the inflow measurement sensor so severely that an on-site repair was not feasible anymore. A follow-up attempt led to another crash that damaged the remaining functionality of the kite sensor unit, so
no further use of the onboard sensor system was possible. As consequence, the second measurement run provided data only up to a power setting of $u_p = 0.7$ and without inflow data. The Prandtl probe was not mounted in the second measurement run. The




steering gain and the dead time were calculated from the GNSS velocity and are plotted in Fig. 23 with the corresponding data of the first measurement run. It is obvious that the kite behaviour differs substantially in the two runs. In the second run, the kite

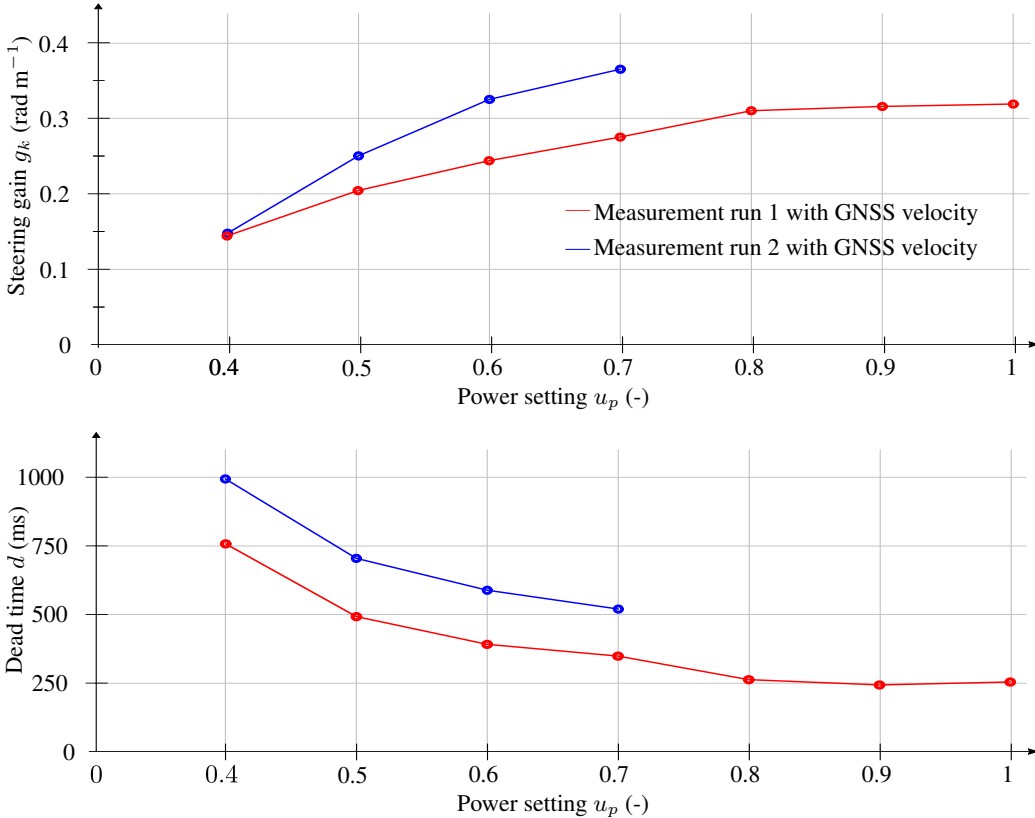

**Figure 23.** Comparison of the mean values of steering gain $g_k$ and dead time $d$ averaged over both tow directions as functions of the power setting $u_p$ for the Vegas 2015 with 10 m$^2$ when performing the MLE with the GNSS velocity. Adapted from Elfert (2021).

is overall more agile, reacting faster and stronger to control inputs. The deviations can be explained by comparing the velocities
and line forces during the two runs. The simulated wind and GNSS velocities of representative data sections are illustrated in Figure 24. The mean values differ only slightly between the two runs, with the simulated wind velocity about 0.5 m s$^{-1}$ higher in the first run and the GNSS velocity about 0.3 m s$^{-1}$ higher in the second run. Using the GNSS velocity as a measure of the inflow speed, this difference would correspond to an approximate 6%-deviation in aerodynamic forces. However, the line forces plotted in Fig. 25 show that the mean force in the power line is about 30% higher in the second run. With higher
line tension, the kite reacts faster and stronger to control input. The lower line forces in the first run can be explained by the stronger fluctuations of the simulated wind speed during the first run, as seen in Fig. 23. These fluctuations negatively impact

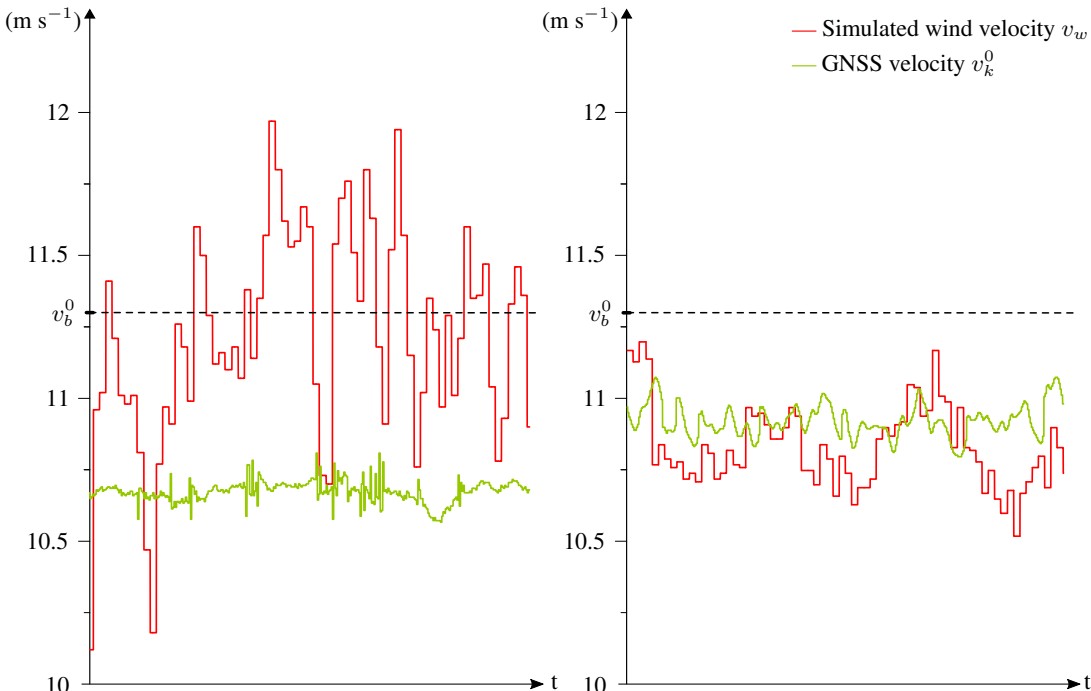

**Figure 24.** Representative cut-outs of the simulated wind and GNSS velocity data during a period of equal length in the first (left) and second (right) measurement run. Tow speed set to $v_b^0 = 11.3$ m s$^{-1}$. Adapted from Elfert (2021).

the aerodynamic performance of the kite, thus decreasing the generated line forces. A thorough computational analysis of the effect of inflow turbulence on the energy harvesting performance of kites was presented in Fechner and Schmehl (2018).

### 5.5   Comparison with literature

In the final step, we compared the measured turning behaviour with experimental results from the literature. In general, the definition of the steering input $\delta$ influences the calculated steering gain $g_k$. However, all studies with measured steering gain also included the steering input in a standardised way, allowing for a direct comparison of the measurements. Because of the linearity of the turn rate law, larger values of the maximum steering line length difference $\Delta l_{SL,max}$ only lead to correspondingly larger amplitudes of the measured yaw angle change $\Delta \psi$, not affecting the gain. Table 3 lists the results of other experimental studies

together with our results.

The derivation of the turn rate law leading to Equation (12) shows that the steering gain $g_k = C/m$ implicitly includes a dependency on the surface-to-mass ratio of the kite because the constant $C$ also includes the effective vertical wing surface area. Since the mass of a kite increases faster than its wing surface area, the steering gain generally decreases with the size of the kite. This is well reflected by the data in Table 3, however, it is also obvious that within a certain size interval, the type of

kite and its specific design can also influence the turning behaviour substantially.



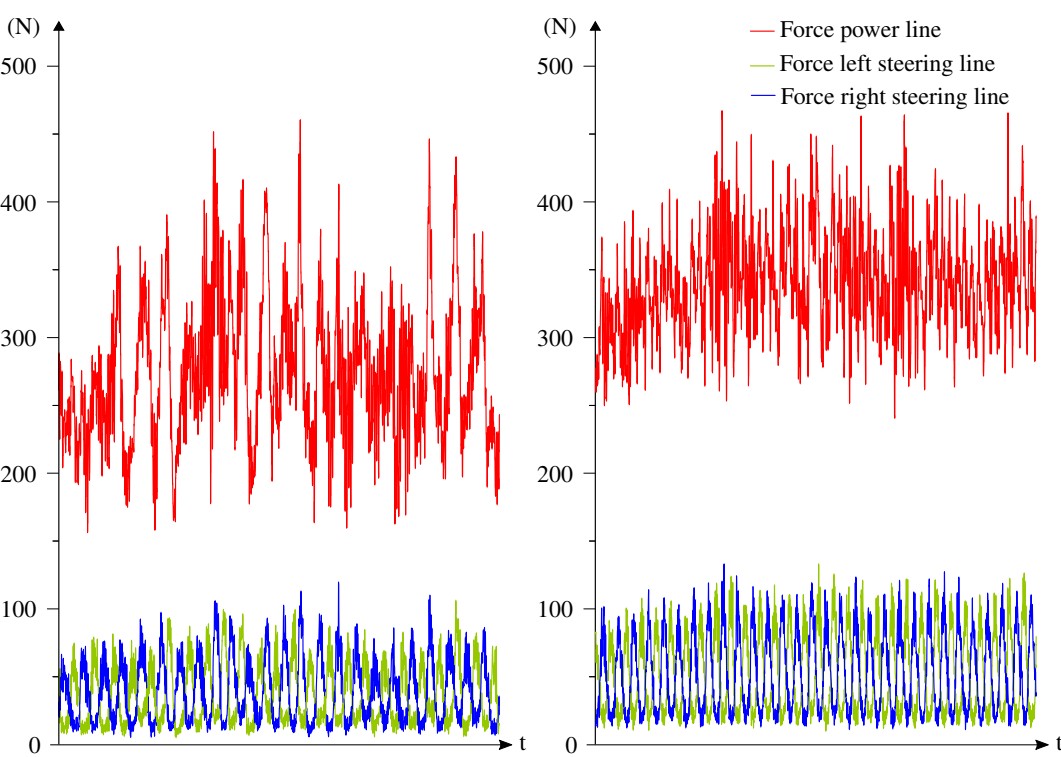

**Figure 25.** Representative cut-outs of the line force data during the periods shown in Fig. 24 in the first (left) and second (right) measurement run. Adapted from Elfert (2021).

## 6 Conclusions

Compared to conventional aircraft, flexible-membrane wings are highly manoeuvrable because of the low inertia and the entire canopy acting as a morphing aerodynamic control surface. The aim of the present study was to expand on an existing tow test setup to systematically quantify this manoeuvrability. For this purpose, a control algorithm from the literature was adapted to

operate the kite in automatic crosswind manoeuvres while towing it at constant speed along a straight runway. The widely used turn rate law was adopted to identify the steering gain and dead time from the measured flight dynamic response to a prescribed steering input. Towing the kite in concatenated back-and-forth loops made it possible to determine the manoeuvrability for the entire range of power settings within a single non-stop measurement run.

     The results confirm the well-known behaviour of increasing agility with the power setting, as evidenced by increasing steer-

ing gain and decreasing dead time. An ambient wind speed leads to deviations for the two opposing tow directions, and the larger the ambient wind speeds, the larger generally the differences in steering behaviour. An extreme effect was observed during the first measurement run when a sudden peak in the inflow speed, potentially due to thermals above a neighbour-





**Table 3.** Measured steering gain and dead time of kites, with and without suspended kite control unit (KCU).

| Experiment | Kite size[a] (m$^2$) | Kite type | KCU | Steering gain (rad m$^{-1}$) | Dead time (ms) |
|---|---|---|---|---|---|
| Erhard and Strauch (2013b) | 160 | ram air | yes | 0.039 | - |
| Roullier (2020), TU Delft V3 | 25 | LEI | yes | 0.17 | - |
| Oehler et al. (2018), TU Delft V3 | 25 | LEI | yes | 0.12 to 0.22 | - |
| Erhard and Strauch (2013a) | 20 | ram air | yes | 0.13 | - |
| Cadalen et al. (2018) | 15 | LEI | no | 0.17 | - |
| Oehler et al. (2018), Genetrix Hydra V5 | 14 | LEI | no | 0.25 to 0.35 | - |
| This study, measurement run 1 | 10 | LEI | no | 0.14 to 0.32 | 480 to 1033 |
| This study, measurement run 2 | 10 | LEI | no | 0.15 to 0.37 | 520 to 994 |
| Costello et al. (2018), Flysurfer Viron | 2.5 | ram air | no | 1.1 | 260 |

[a]Flat, laid out wing surface area

ing cornfield, triggered an anomaly of the flight controller, which, as it stayed undetected, led to a degraded measurement at the specific flight condition. Comparing the first and second measurement runs, the measurements also revealed a substantial impact of turbulent fluctuations of the simulated wind speed on the manoeuvrability of the wing. We concluded that an increasing degree of turbulence has a deteriorating influence on the flow around the wing, lowering the line forces and, in turn, the manoeuvrability.

A key finding of the present study is thus that the measurement quality crucially depends on the ambient wind conditions, which must be monitored closely. The experimental campaign revealed several practical limitations of the tow test setup. Firstly, a Prandtl probe protruding from the wing's leading edge is a vulnerable high-precision instrument, especially when considering that flexible wings flying fast crosswind manoeuvres in a natural wind environment close to the ground are prone to occasional crashes. By design and nature, flexible wings cannot ensure rigorous flight stability as fixed-wing aircraft can. The loss of the probe and the entire onboard sensor module at the start of the second measurement run severely impaired the potential output of the measurement campaign. However, we also found that for sufficiently low ambient wind speeds, the flight velocity of the kite relative to the ground, denoted as GNSS velocity in the present work, can be used as a reliable indirect measure of the inflow velocity.

Secondly, operating a wing in crosswind manoeuvres greatly amplifies the pulling force. For safety purposes, we could tow only relatively small wings, flying these close to the edge of the wind window at relatively high elevation angles (around 70°) where complete figure-of-eight loops were often not feasible. For the larger kites used in airborne wind energy applications, it appears more adequate to identify the steering gain and dead time based on operational data in the original application setting with the fixed ground station. However, the effort to integrate a new kite design into the complete system hardware and control system can be considerable. Using the described setup and procedure for tow tests of a scale model of the wing may be the more efficient way to evaluate new kite designs.



*Code and data availability.* The code and measurement data can be made available in the framework of a cooperation agreement.

*Author contributions.* CE designed the test setup with the support of DG and performed the measurements. The preparation of the paper was the joint work of CE, DG and RS.

*Competing interests.* At least one of the (co-)authors is a member of the editorial board of *Wind Energy Science*.

*Acknowledgements.* The authors would like to thank Jelle Poland for providing feedback on the manuscript.

*Financial support.* RS has received funding from the European Union Horizon 2020 research and innovation program under Marie Skłodowska-Curie grant agreement no. 642682 for the ITN project AWESCO and grant agreement no. 691173 for the "Fast Track to Innovation" project REACH.



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
