# Peer review of "Measurement of the turning behaviour of tethered membrane wings using automated flight manoeuvres"

_Wind Energy Science, 2024_

## Author Response (AR1)

Manuscript:   WES-2024-87 | Research article
Submitted:    14 Jul 2024
Title:        **Measurement of the turning behaviour of tethered membrane wings using automated flight manoeuvres**
Authors:      Christoph Elfert, Dietmar Göhlich, and Roland Schmehl

**Reviewer 1**

The paper describes a test setup to evaluate the steering behavior of soft kites. Although the concepts are definitely not new, as shown by the relatively high number of previous work reporting similar findings that the authors correctly mention, the experimental setup and test procedures are very well described and of extremely high value to researchers in the field of airborne wind energy.

I recommend acceptance as it is

Response: Thank you very much for taking the time to review out paper and accepting is as is.

**Reviewer 2**

The paper presents a test procedure to measure the steering behaviour of tethered membrane wings (kite). The main concept and the turn rate law modelling are supported by established works in literature. The measurement procedure was designed to collect sufficient data over the range of wind power densities using automatic flight manoeuvres under controlled environmental conditions. Two sensors were built and used to take measurements from the kite and from the test bench. A control system was adapted to operate the kite in automatic crosswind manoeuvres while being towed at constant speed along a straight runway. The experimental testing and analysis are comprehensive.

The paper is well written, can be accepted as it is.

Specific comments

- In the test, how to implement the incremental change of power setting $u_p$? I also didn't follow how the change of steering input was implemented, by changing $\alpha_{bar}$ or what?
- Some section titles could be expanded to show more of the main work/contribution of the associated section, for example, Section 3 covers the two different sensing systems, Section 4 includes the control algorithm.
- Since a large number of experimental data have been collected under controlled operations, for the future work, it may be possible to include the power setting parameter explicitly into the model for the steering gain and the deadtime.

Response: Response: Thank you very much for taking the time to review our paper and accepting is as is.

Specific comments

- In the test, how to implement the incremental change of power setting $u_p$? I also didn't follow how the change of steering input was implemented, by changing $\alpha_{bar}$ or what?

Response: We realize that an important detail in the test schematic in Figure 2 was missing (two line segments connecting the control bar with the pivot unit on the test bench) and that the

accompanying description of the steering actuation from pilot to kite was unclear. We updated the schematic and clarified the text.

To summarize: the pilot uses live video of the kite on a display in the towing vehicle and a conventional three-line control bar (customary in kite sports) to generate the steering commands. These steering commands are converted by servo motors into digital signals, which are transmitted to the test bench. On the test bench, the signals are used to actuate the two steering lines, recreating the manual line actuation of the pilot in the towing vehicle. The steering and power lines of the kite connect to the pivot unit via a second control bar. This control bar has the same geometry as the bar that the pilot uses. The actuation on the test bench and in the towing vehicle are mechanically separated, only communicating via digital data transfer. This complete mechanical separation allows for replacing the manually generated steering commands from the pilot by an autopilot system. Below, you will find CAD renderings of the pivot head on the test bench (left) and the pilot interface in the towing vehicle (right).

[Figure]

Pivot head on test bench                    Pilot interface in towing vehicle

Images from: Elfert, C.: Methodische Untersuchung des dynamischen Verhaltens seilgebundener, hochflexibler Tragflächen auf Basis automatisierter Flugmanöver, Ph.D. thesis, Technische Universität Berlin, https://doi.org/10.14279/depositonce-11848, 2021

- Some section titles could be expanded to show more of the main work/contribution of the associated section; for example, Section 3 covers the two different sensing systems, and Section 4 includes the control algorithm.

Response: Good point. We adjusted the subsection titles in Section 3 to:

3.2 Measuring kite position, velocity and orientation

3.3 Measuring relative flow velocity at the kite

- Since a large number of experimental data have been collected under controlled operations, for the future work, it may be possible to include the power setting parameter explicitly into the model for the steering gain and the deadtime.

Response: Good point. Including the power setting explicitly in the model for the steering gain and the deadtime should be straightforward with a purely data-driven approach (surrogate model). Accounting for the underlying physics (mechanistic model) will be more challenging because changing the power setting of a flying C-shaped kite leads to a rather complex aerostructural response of the tensile membrane structure. While the pitch of the wing is changing, the wing is also deforming. This deformation depends on the geometry of the wing and the bridle line system. In a parallel study (Poland & Schmehl: "Modelling Aero-Structural Deformation of Flexible Membrane Kites". Energies, 2023. https://doi.org/10.3390/en16145264), we developed a simple mechanistic model to describe the induced geometry change of a C-shaped kite when depowering. This could be a starting point for a mechanistic model.

**Reviewer 3**

The paper describes a test procedure that can be used to identify and assess the turning performance of flexible membrane kites.

While there already exists a good amount of previous work on performance test procedures for flexible kites, this is the first to investigate in particular the turning behavior in a controlled environment, which is a complex task to complete in the real world.

This is achieved by towing the kite on a straight runway during low ambient wind conditions, in combination with an automatic flight controller to perform the figure-eight maneuvres.

On the conceptual side, the work successfully combines sensor fusion approaches from the literature (to obtain turning rate estimates from the measurements) with existing modeling approaches to estimate a linear steering-to-turning-rate model with dead-time for different operational settings.

The experimental flight maneuvres are obtained by tailoring an automatic control method from the literature to the test procedure at hand.

The numerical results presented show the efficacy of the proposed approach, while anomalies in the data and the encountered challenges are explicitly addressed.

Overall, the proposed methodology and the showcased knowledge necessary for real-world application is of very great value to the airborne wind energy community.

The technical details of the procedure are presented comprehensively and motivated in great detail.

The results are discussed critically, including a sensitivity analysis of the post-processing approach and a comparison and validation of the outcome against results from literature.

The text is written very well, including many helpful and clean visualizations.

In conclusion, my recommendation is to accept the paper as is.

Response: Thank you very much for taking the time to review out paper and accepting is as is.